# Geospatial assessment of household water, sanitation and hygiene conditions and associated factors in Nigeria: A causal relationship model

Jacob W. Mobolaji[1], Akinola Shola Akinwumiju[2]*

1 Department of Demography and Social Statistics, Obafemi Awolowo University, Ile-Ife, Osun State, Nigeria, 2 Faculty of Science and Engineering, University of Wolverhampton, Wolverhampton, West Midlands, United Kingdom

☯ These authors contributed equally to this work.
* a.akinwumiju@wlv.ac.uk, akinshola822@gmail.com

## Abstract

Lack of adequate access to safe water, sanitation, and hygiene (WASH) has contributed to increased under-five mortality and morbidity of school-age children in low- and middle-income countries. Despite the global and national intervention programs, access to safe WASH remains a critical challenge in Nigeria. This study employed spatial and non-spatial statistics to establish causal relationships between WASH conditions and household factors in Nigeria. Results show that a large proportion of Nigerian households were still associated with unimproved hygiene (88%), sanitation (47%) and water (25%). Wealth status, literacy level and residency type exhibit significant causal relationships with households' water sources ($\alpha = 0.000$). Wealth status and the gender of household head exhibit significant causal relationships with sanitation condition ($\alpha = 0.000$) and hygiene condition ($\alpha = 0.004$ and $\alpha = 0.345$, respectively). However, the computed parameter Degree of Dependence (DoD_j) shows that the choice of water sources mostly depends on residency type (DoD_j = 0.998) compared with the level of education and wealth status (DoD_j = 0.535 and 0.485, respectively). Statistical indices show that the implemented regression models are reliable (with models' DoD of 0.714–0.996, Adjusted $R^2$ of 0.184–0.762 and Akaike Information Criterion (AICc) of 68–103). The study concludes that a high risk of unimproved WASH is associated with rural residence, which is usually characterised by a low level of education, poverty and large household size. It further concludes that the high prevalence of unimproved hygiene, irrespective of the household wealth status and educational level, suggests the need for proper health and hygiene education. This study suggests the need for a more focused policy action towards empowering rural and vulnerable households in Nigeria with relevant preventive environmental and health information and appropriate social support for the communities.

**Data availability statement:** All Datasets are freely available on: https://osf.io/ehkav. Demographic and Health Survey (DHS) Data are available at: https://www.dhsprogram.com/data/dataset/Nigeria_Standard-DHS_2018.cfm?flag=0.

**Funding:** The author(s) received no specific funding for this work.

**Competing interests:** The authors have declared that no competing interests exist.

## Introduction

Water, Sanitation, and Hygiene (WASH) play an essential role in public health, particularly in low- and middle-income countries (LMICs). According to the World Health Organisation (WHO) and United Nations Children's Fund (UNICEF) Joint Monitoring Programme, access to drinking water refers to not only the availability, but also the accessibility and quality of household's main water source for domestic uses including drinking, cooking, and personal hygiene [1]. Sanitation refers to the ability to utilise sanitation materials designed to hygienically treat and dispose of human faeces off-site or channelled through a sewer system with wastewater [2]. Hygiene is the health-maintaining conditions and practices that help to prevent the spread of diseases and infections, including handwashing and hygienic food preparation, and dirt management [3]. Adequate access to clean water, improved sanitation facilities, and hygienic practices are fundamental to reducing the burden of waterborne diseases, under-five mortality, and improving overall health outcomes [4]. In 2018, about 820,000 diarrhoea-induced deaths were associated with unsafe WASH, with more elevated impacts on the under-five children [5]. Children from LMIC, including Nigeria, have the largest share of the death toll; yet, with access to safe sanitation and hygiene practices, 95% of the under-five deaths are preventable [6].

Despite global efforts toward improving WASH conditions, Nigeria continues to face significant challenges in ensuring equitable access to these basic facilities, particularly in rural and underserved communities [7–8]. The United Nations, through the Sustainable Development Goals (SDGs) 6, set a 15-year target to ensure sustainable management of water and sanitation for all. While reasonable progress has been recorded by some countries [9], the majority of LMICs, including Nigeria, are lagging. According to a 2020 report, a third of Nigerians lack access to basic water, with over 46 million using open defecation and 167 million without hygiene facilities [10]. Although, the government has embarked on various policies and programmes to address the menace, the progress recorded has been marginal, and as of 2021, a substantial proportion of Nigerians are still deprived of access to basic water (33%), sanitation (54%) and hygiene (82%) [11]. The government's proposed obliteration of open defecation by 2025 does not seem attainable due to its over-concentration on public institutions with little attention to the situations in the households. This has contributed to the country's slow progress in achieving Sustainable Development Goal (SDG) 6 of ensuring availability and sustainable management of water and sanitation for all by 2030.

Nigeria's WASH sector is characterised by widespread subnational and rural-urban inequities. For instance, rural areas (43%) are more affected by the unavailability of improved water sources compared to urban areas (24%) [12]. At the subnational level, studies have shown that northern Nigeria is disproportionately affected by inadequate access to water and sanitation services, largely due to the arid climate, conflict, and socioeconomic challenges [13]. Stressing the deplorable state of sanitation facilities, in particular, the open defecation practice is more prevalent in northern Nigeria than in any other region [14]. Southern Nigeria, by contrast,

has relatively better WASH conditions but still faces challenges in peri-urban and slum areas [15,16]. These disparities have been linked to high rates of cholera and diarrheal diseases, undernutrition, and mortality, especially among under-five children in northern Nigeria [17,18].

Geographic and systemic factors such as the political landscape, government policies, and the strength of local institutions play important roles in access to WASH in LMICs. In Nigeria, weak governance and corruption have been identified as barriers to the effective implementation of WASH programs in Nigeria [14]. These barriers manifest in lack of political will and inadequate funding, resulting in poor maintenance of infrastructure, limited access to safe water and sanitation in targeted institutions and communities [19]. Environmental factors, including the climate, physical geography and environmental degradation, also impact WASH programmes and access. In northern Nigeria, the semi-arid and arid climate contributes to water scarcity, making it difficult for households to access safe drinking water [20]. Whereas, while water is available in the southern regions, water quality is often compromised by flooding and poor waste management, leading to contamination of drinking water sources [21].

However, the geographic and systemic factors influencing access to WASH in LMICs intersect with socioeconomic factors. Evidence from other countries suggests that a lack of access to safe drinking water is associated with individual and household peculiarities. For instance, in Malawi, White et al. [22] associated poverty, limited education, urban residence and being a female with increased limitation to accessing improved water, sanitation and hygiene [22]. The available evidence in Nigeria is scant. Although a study identified poverty as a major factor influencing access to WASH among semi-urban residents in Nigeria [23], this evidence is limited and not nationally representative, undermining the country's diverse geographical and socioeconomic landscapes with varied WASH intervention programmes. Besides, the evidence is based on individuals. The household characteristics potentially impacting localised access variations are yet to be explored. Our study argues that evidence on access to safe WASH from a household perspective gives a better picture of the WASH realities of all individuals in the household, and serves as a reference for informed policy.

In addition, while WASH conditions have been extensively studied in LMICs, the use of geographic information systems (GIS), which is highly beneficial, has been underutilised. Specifically, research focus has been on Africa and Asia-Pacific, due to the obvious low standard of living [24–31]. Scientific research in this area has benefited from data obtained from both primary and secondary field surveys, and the employment of traditional statistical tools for data analyses has been commonplace over time. On very rare occasions, geographic information systems (GIS) have been employed to visualise the results of statistical analyses. In most cases, scientific studies on WASH conditions are monotonous with emphasis on conventional data analyses and interpretation. Geospatial technologies such as GIS and remote sensing have become increasingly popular tools for assessing and identifying spatial patterns of inequality in environmental issues [32]. Such innovations could provide valuable information on the role of location factors, and the information could enhance the effectiveness of relevant interventions, particularly in data-scarce and resource-poor regions. These tools can be used to highlight variations in WASH accessibility, coverage, and practices across different geographies of Nigeria. By integrating geospatial data with Nigeria's household demographic and socioeconomic information, we can better understand the drivers and identify areas where interventions are most needed. The geographical and regional differences in Nigeria underscore the need for targeted interventions based on geospatial evidence. Given the country's rapid population growth and the widespread WASH inequalities, innovative approaches are needed to monitor and improve the conditions. The adoption of geospatial analysis in this study will enable more accurate mapping and assessment at household-level situations, providing insights into spatial disparities and associated factors in Nigeria. Thus, the main objective of this study is to examine the relationship between household factors and WASH conditions. The study also aims to evaluate the importance of spatial context and model the spatial pattern of household factors and WASH conditions in Nigeria.

## Materials and methods

### The study area: Nigeria

Nigeria is the most populous black nation in the world. The country is located north of the Equator and east of the Greenwich Meridian with a centroid coordinate of 9°04'39.90" N and 8°40'38.84" E (Fig 1). As of 2022, the official projected population of Nigeria was 216,783,381, of which 108,432,971 were female [33]. In 2022, 55% of Nigerians were in the age group 15–64, while 42% and 3% were in the age groups 0–14 and 65+, respectively. Nigeria accounts for 15.2% of Africa's population and has the largest population of black women and children in the world. The country is subdivided into six geo-political zones – North-East (six states), North-West (eight states), North-Central (five states), South-East (five states), South-West (six states) and South-South (six states).

### Ethics approval and consent to participate

This study utilised secondary data from the Demographic and Health Survey (DHS) program being coordinated by ICF International. To utilise the data for our study, written approval was obtained from ICF International. The DHS protocol was approved by the National Health Research Ethics Committee of Nigeria (NHREC) and the ICF Institutional Review Board (IRB). The IRB-approved procedures for the DHS public-use datasets do not in any way identify the participating respondents, households, or sample communities.

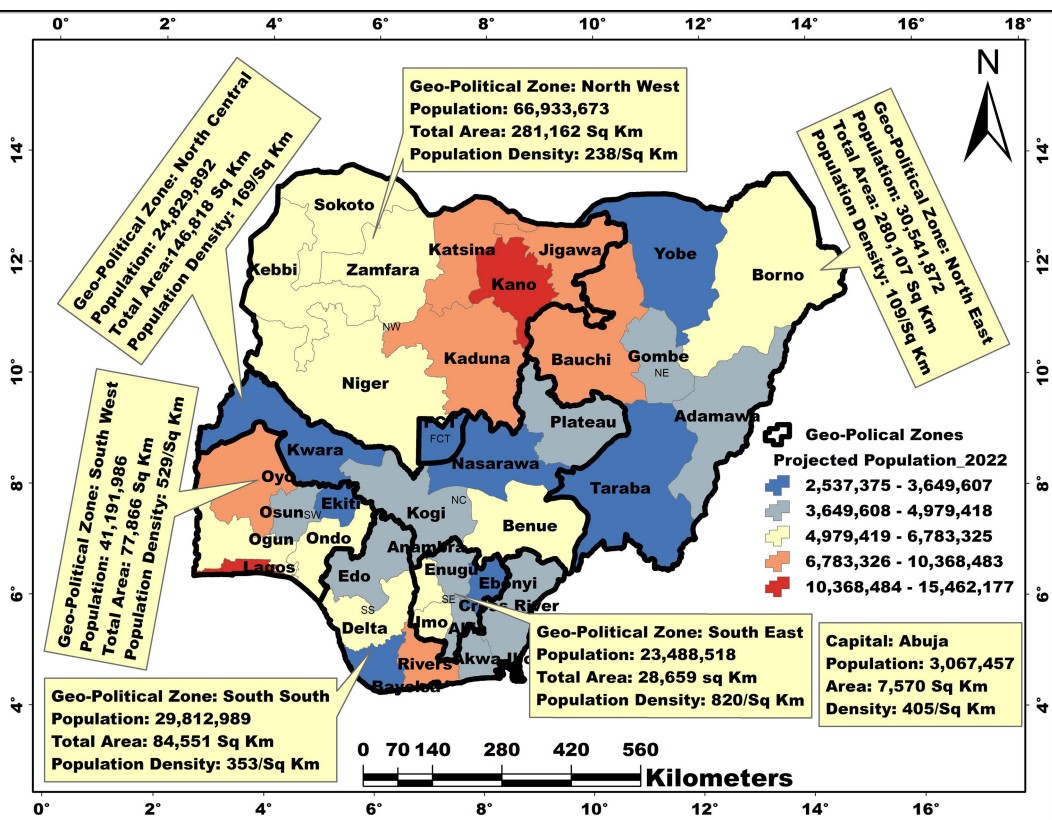

**Fig 1. Map of Nigeria showing population distribution by states and geo-political zones based on National Population Commission Projection, 2022.** This map was created on the ArcGIS 10.8 platform.

## Data sources and sample design

The study utilised the household dataset of the 2018 Nigeria Demographic and Health Survey (DHS). The dataset elicits demographic, environmental and health information about the sampled households across the country. The households were selected using a stratified multi-stage cluster design. The details of the sample design and data collection methods were published in the DHS report [12]. This study was based on the households, not on the individual household members. A weighted total sample of 40,369 households in the datasets was analysed for this study. Since the missing values were minimal (<1%), listwise deletion was used to deal with the missing values.

## Variable measurement

In this study, the outcome variables are the unimproved water (UW), unimproved sanitation (US) and unimproved hygiene (UH) generated from household sources of drinking water, type of sanitation and type of hygiene, respectively. On the sources of drinking water, responses of the household head to the question, "what is the main source of drinking water for members of your household?" were categorised into two: improved water sources (water piped into the house, yard/ plot or to neighbour, standpipe or public tap, tube well or borehole, protected well and springs, rainwater, water obtained via a tanker truck or cart with small tank, bottled water and sachet water) and unimproved water sources (unprotected well and springs, and river/dam/lake/stream water). The type of sanitation was measured with the question "what kind of toilet facility do members of your household usually use?" and the responses were categorised into two: improved sanitation (use of flush or pour-flush toilets that flush waste and water to piped sewer system, septic tank or any other closed pit latrine, ventilated improved pit latrine or pit latrine with slab, toilets with an unknown destination and composting toilets) and unimproved sanitation (use of pit latrine without slab, open pit, bush or field, bucket toilet and hanging toilet or latrine). Handwashing and location for cooking food were used to measure household hygiene. While the interviewers observed the availability of a place, water and soap detergent or any other cleansing agents for handwashing, the question "Is the cooking usually done in the house, in a separate building, or outdoors?" was used to assess the location for cooking household food. The type of hygiene was categorised into two: improved hygiene if handwashing place, water and soap are observed and household cooking is done indoors; and unimproved hygiene if otherwise in any of the two.

The independent variables are the characteristics of the households, including age and sex of the household head, household size, household wealth quintile (poorest, poorer, middle, richer and richest), household educational level (low, middle and high), and rural-urban place of residence. While all other variables were in the dataset, household educational level was computed based on the proportion of the household with the various levels of education in the household. Each household member was scored based on the respective level of education. Household members with higher education were scored 5 points, while members with secondary or equivalent education were scored 4 points, primary education as 2 points, and no formal education as 1 point. We obtained the mean household education score by dividing the total household education score by the number of household members. The mean scores were divided into tertiles such that the top third was categorised as high, the middle third as medium, and the bottom third as low.

## Data analysis

First, we employed univariate statistical analysis (i.e., frequency and percentage distributions) to explore the household characteristics and the proportion of households with unimproved water, sanitation and hygiene in each state in percentages. The analyses were done using Stata version 15.1 [33]. Ultimately, we implemented GIS-based global and local regression models to examine the causal relationship between household factors and WASH conditions. We equally adopted spatial statistical modelling algorithms to map the proportional patterns of WASH conditions and household factors in Nigeria. The adopted methods are further described below. We believe that the adoption of spatial statistics will provide a unique opportunity to explore, model and evaluate the spatial variability of the causal relationships that exist between household factors and WASH conditions.

## Spatial regression analysis

**Global regression models.** The Ordinary Least Squares (OLS) Regression Model assumes that there is no interdependence among parameter values. Thus, it is believed that both response and supposed predictor variables independently vary in space and time. In deviation from this assumption, the interrelationship among spatial phenomena and events interwovenly varies in space and time, hence the existence of spatial autocorrelation. Thus, OLS is known to be unsuitable for modelling location-specific association among spatial phenomena and events [34,35].

**Ordinary least squares regression.** OLS is defined as:

$$y_i = \beta_0 + x_1\beta_1 + x_2\beta_2 + x_3\beta_3 + x_i\beta_i + \varepsilon_i \tag{1}$$

Where,

$y_i$ is the response variable, $x_1$, $x_2$, $x_3$, $\ldots x_i$ are the explanatory variables, $\beta_0$ is the intercept, $\beta_1 \ldots \ldots \beta_i$ are the partial regression coefficients and $\varepsilon_i$ is the error term. Here, the WASH conditions are the response variables, while the predictor variables consist of the household parameters. For OLS in particular, β is optimised by minimising the sum of prediction errors of squared [34,35]. Usually, OLS is validated by the supposed non-existence of interdependence among variable counts that are assumed to be fixed in space. In the same vein, error terms are not supposed to be correlated [34–36]. *Local regression model:* The Global regression models (e.g., Ordinary Least Squares (OLS), Spatial Error Model (SEM), and Spatial Lag Models (SLM) generally assume that a response variable will engage in unvarying relationships with the explanatory variables over the whole study site [35,37,38]. However, the assumption of a constant relationship is unrecognisable when phenomena, locations and neighbourhood criteria are prioritised. To accommodate the concept of spatial variability in regression models, local regression models (such as Geographically Weighted Regression (GWR) and Multiscale Geographically Weighted Regression (MGWR)) were proposed based on kernel-weighted regression [37]. Local regression models were introduced to inculcate location-specific variability in spatial interactions. Instead of the globalised estimation of parameter values, local regression models permit isolated location-specific estimation of regression parameters for individual entities (i.e., states or local government areas), thereby introducing the context of spatial variability into the regression model [35,36]. Specifically, MGWR recognises spatial variability of parameter relationships as a precondition in spatial regression analysis. It can analyse these relationships locally, based on the possibility of accommodating multiple bandwidths over a study area. Unlike the previous regression models that assume a unified, homogeneous bandwidth, MGWR and its clones assume that the spatial relationship varies from one location to another over a geographic space [35,39].

**Multiscale geographically weighted regression.** MGWR is defined as [35,39]:

$$y_i = + \sum_{j=1}^{m} \beta_{bwj}x_{ij} + \varepsilon_i \quad i = 1, 2, \ldots, n \tag{2}$$

where $\beta_{bwj}$ represents the chosen bandwidth with which the j$^{th}$ relationship is calibrated [39], while other parameters remain as in Eq. (2). MGWR is regarded as a generalised additive model (GAM), admitting the adoption of backfitting algorithms for its calibration [35,39].

As a GAM, MGWR is redefined as:

$$y_i = \sum_{j=0}^{m} f_{ij} + \varepsilon \tag{3}$$

where $f_{ij}$ replacing $\beta_{bwj}x_{ij}$ in equation (3) denotes the j$^{th}$ additive term denotes an applied smoothing function to the j$^{th}$ predictor variable at state $i$ [35,36,39]. Thus, a separate bandwidth is established for individual j explanatory variables in the process of model calibration [35,36,39].

In our study, the spatial entities are the 36 states of Nigeria and its federal capital territory. The dependency modelling hinges on a local regression that is characterised by a predetermined bandwidth that changes at specific geographical scales, regardless of the state's size. Thereby, providing the opportunity to analyse and compare individual bandwidths [36]. In order to create the spatial perspective required for the implementation of MGWR, the centroid coordinates of the studied spatial entities were extracted and converted to point features. Thereafter, multiple parameter values (representing household parameters and WASH conditions) were extracted into the spatial database of the earlier created point features to generate input datasets for the MGWR model. The examined parameters are: the household parameters (household head aged 60 + years (HHA60Y), female-headed households (FHH), households below middle wealth quintile (HBMWQ), households with low level of education (HWLLE), households with 6 + members (HW6M), households with rural residence (HRR)) and the WASH conditions (unimproved water (UW), unimproved sanitation (US), unimproved hygiene (UH) to create a relationship, where the household parameters constitute the explanatory variables and the WASH conditions are the response variables. The database of the point features was subjected to spatial regression analyses on the MGWR 2.2 platform, and the outputs were joined to the database of the studied spatial entities on the ArcGIS platform.

## Analytical procedure

In this study, we employed exploratory regression analysis to isolate combinations of household factors that have a significant influence on the individual WASH conditions – unimproved water, unimproved sanitation and unimproved hygiene. In this case, the WASH conditions are the response variables, and the household factors were passed into the model as possible explanatory variables. The outputs of the exploratory regression analyses reveal that WASH conditions are influenced by only four (i.e., HMWQ, HLLE, HRR and FHH) variables in Nigeria. To be specific, UW is influenced by HMWQ, HLLE and HRR, while US is influenced by HMWQ and FHH; and UH is influenced by FHH and HMWQ.

We adopt OLS (global) and MGWR (local) regression models to examine the relationships between the WASH conditions and the four isolated household factors. In this case, the WASH conditions constitute the response variables while the household factors are the explanatory variables. In the recent past, spatial regression models have been widely employed in medical geography and spatial epidemiological studies [35,39,40]. Recent studies have shown that MGWR and its clones are highly suitable for modelling spatial relationships and location-specific parameter influence. Like OLS, MGWR can define the direction of the relationship that exists between a response variable and individual predictor variables [35,36]. Despite its shortcomings, OLS is an efficient base model for spatial regression analysis. It gives an insight into the possible direction of the relationship that exists between spatial phenomena. To start with, OLS was adopted to regress the WASH conditions (i.e., response variable) on the isolated household factors (i.e., explanatory variables). The OLS-derived variance inflation factor (VIF) and strong probability (b) revealed that a spatial regression model for predicting the WASH conditions could be developed on HMWQ, HLLE, HRR and FHH. In the case of UW, two passing models were identified – HMWQ + HRR and HLLE + HRR. As for US and UH, the passing models could only be built on HMWQ + FHH and FHH + HMWQ, respectively. In this case, only two-parameter models were adjudged to pass the multicollinearity test. Thus, the variability of WASH conditions is influenced by only four household factors in Nigeria. In this study, we executed OLS on ArcGIS 10.8 (ESRI.com) and MGWR 2.2.1 (https://sgsup.asu.edu/sparc/mgwr) platforms, while MGWR was solely implemented on the MGWR 2.2.1 platform. To calibrate our MGWR model, we chose an adaptive bi-square as the spatial kernel, and Gaussian was selected as the model type, while the Akaike Information Criterion was preferred as an optimisation criterion. The performances of the regression models were judged on the basis of adjusted $R^2$ and Akaike Information Criterion (AICc). We geo-visualised the spatial patterns of the coefficient of determination in order to gain a spatial perspective of the influence of household factors on WASH conditions over Nigeria.

Furthermore, the WASH conditions and the influencing household factors were subjected to thematic cartograms modelling. A cartogram is a spatial descriptive statistical model that portrays data distribution proportions about sampled spatial locations. In addition, the WASH conditions and the influencing household factors were subjected to scatter plot matrix

and box plot algorithms. The parameters' cross plots, histograms, correlations and the accompanied significant levels are provided by the scatter plot matrix. The box plot visualises the internal structures of the examined parameters. In addition, it also provides basic descriptive statistics including the mean, median and interquartile range. The spatial descriptive analyses were implemented on the GeoDa 1.18 platform (geodacenter.github.io).

## Results

### Household characteristics

The results in Table 1 indicate that more than three-quarters (79.3%) of the households were headed by persons below age 60 years. This indicates that this proportion is still in the labour force age classification, while the rest were outside the economically active age. At least 4 in 5 of the households were headed by a male; about 36% were below the middle wealth quintile; and the majority (78%) had less than 6 household members. Overall, more than half (53%) of the households reside in rural areas, with about a quarter in the North-West, a fifth in the South-West, and 12–14% in other regions.

**Table 1. Background characteristics of the households in Nigeria.**

| Characteristics of the respondents | Frequency (N = 40,369) | Percentage |
|---|---|---|
| Age of household head | | |
| <60 | 31,992 | 79.3 |
| 60+ | 8377 | 20.7 |
| Sex of the household head | | |
| Male | 33088 | 82.0 |
| Female | 7281 | 18.0 |
| Household wealth status | | |
| Poorest | 6903 | 17.1 |
| Poorer | 7485 | 18.5 |
| Middle | 8262 | 20.5 |
| Richer | 8666 | 21.5 |
| Richest | 9053 | 22.4 |
| Household level of education | | |
| Low | 13585 | 33.7 |
| Middle | 13432 | 33.3 |
| High | 13352 | 33.0 |
| Household size | | |
| 6 or fewer | 31505 | 78.0 |
| More than 6 members | 8864 | 22.0 |
| Type of place of residence | | |
| Urban | 18899 | 46.8 |
| Rural | 21470 | 53.2 |
| Region | | |
| North central | 5695 | 14.1 |
| North East | 5694 | 14.1 |
| North West | 9816 | 24.3 |
| South East | 4746 | 11.8 |
| South South | 5715 | 14.2 |
| South West | 8703 | 21.6 |

[a]there are some missing responses.

## Relationship between WASH conditions and household factors

The results of the exploratory regression analyses affirmed that only four out of the surveyed household factors have a significant influence on WASH conditions. Thus, we examined four pairs of global and local regression models. In all, UW was regressed on two pairs of regression models, while US and UH were individually regressed on a pair of regression models. All the examined regression models fulfilled the between variables multicollinearity threshold value of less than 5, which is required for running a spatial regression [35]. Regression algorithms will drop a variable when its variance inflation factor is higher than 5; hence, the reason many of the household factors were left out of the models. The results of these analyses are presented below.

Two predictive regression models were developed for unimproved water. The first model comprised HBMWQ and HWRR, and the second comprised HWLLE and HWRR. For unimproved sanitation and unimproved hygiene, the predictive models comprised HBMWQ and FHH. Results show that unimproved water was positively influenced by HBMWQ, HWLLE and HWRR, while unimproved sanitation and unimproved hygiene were positively influenced by HBMWQ but negatively influenced by FHH. As portrayed by the Joint F-statistic and Joint Wald Statistic, the coefficients of determination for all the OLS models were statistically significant (Table 2). The results reveal that the isolated household factors exerted positive and significant influence on the examined WASH conditions, except FHH, which exerted significant negative but very low influence on unimproved hygiene at $p < 0.01$ (Table 3).

The regression models' goodness of fit is presented in Table 4. The OLS regression model results indicate that HBMWQ.HRR and HLLE.HRR, respectively, could only account for 59% and 55% of unimproved water, which are significant (with $p < 0.01$) and considerably high. In the same vein, HBMWQ.FHH accounts for 54% of unimproved sanitation, whereas FHH.HBMWQ could only account for 20% of unimproved hygiene. The computed AICcs are generally low, ranging from 78.31 (for unimproved water = HBMWQ.HRR) to 103 (for unimproved hygiene = FHH.HBMWQ). However, implementation of a local regression model would guarantee additional model performance as the between-parameters relationships are treated as localised events that do vary in space and time [35].

As expected, MGWR yields better model statistics in the form of higher adjusted $R^2$ and AICc values compared to OLS (Table 4). This is because MGWR takes into cognisance the influence of the inherent spatial variability of the between-parameters relationships (i.e., the uniqueness of the spatial causal effects between household factors and WASH conditions). However, the results show that a lower adjusted $R^2$ and slightly higher AICc values are recorded for unimproved hygiene = FHH.BMWQ MGWR's model which is quite strange. Nevertheless, MGWR model results showcase an overall improvement over that of OLS as depicted by the adjusted $R^2$ and AICc values. In this case, the best individual performance is recorded for MGWR, having the highest adjusted $R^2$ of 0.76 as well as the least AICc value of 68.3. Generally, both OLS and MGWR have managed to account for a substantial percentage of the regression plane. However, the best

**Table 2. OLS diagnostics.**

|  | UW_HMWQ-HRR | UW_HWLLE-HRR | US_HMWQ-FHH | UH_FHH-HMWQ |
|---|---|---|---|---|
| Multiple R-Squared [d]: | 0.621 | 0.584 | 0.570 | 0.252 |
| Adjusted R-Squared [d]: | 0.599 | 0.559 | 0.545 | 0.208 |
| Joint F-Statistic [e]: | 27.913* | 23.850* | 22.564* | 5.726* |
| Joint Wald Statistic [e]: | 35.443* | 41.674* | 85.344* | 21.739* |
| Koenker (BP) Statistic [f]: | 9.241* | 7.126* | 0.643 | 0.218 |
| Jarque-Bera Statistic [g]: | 10.775* | 0.805 | 0.835 | 14.096 |

UW unimproved water; US unimproved sanitation; UH unimproved hygiene; HBMWQ households below middle wealth quintile; FHH female-headed households; HRR households with rural residence; HWLLE households with low level of education. * = significant relationship at p < 0.01

**Table 3. Summary of OLS results – model variables.**

| Variable | Coefficient [a] | StdError | t-Statistic | Probability [b] | Robust_SE | Robust_t | Robust_Pr [b] | VIF [c] |
|---|---|---|---|---|---|---|---|---|
| | | | **UW** | **HBMWQ** | **HRR** | | | |
| Intercept | 4.121 | 4.018 | 1.026 | 0.312 | 2.702 | 1.525 | 0.136 | ----- |
| HBMWQ | 0.365 | 0.081 | 4.517 | <0.001* | 0.093 | 3.942 | <0.000* | 1.610 |
| HWRR | 0.158 | 0.083 | 1.909 | 0.065 | 0.056 | 2.841 | 0.008* | 1.610 |
| UW_HLLE – HWRR | | | | | | | | |
| Intercept | 2.798 | 4.236 | 0.660 | 0.513 | 2.842 | 0.985 | 0.332 | -----. |
| HBMWQ | 0.342 | 0.087 | 3.935 | <0.001* | 0.098 | 3.492 | 0.001* | 1.413 |
| HWRR | 0.216 | 0.081 | 2.648 | 0.012* | 0.073 | 2.972 | 0.005* | 1.413 |
| | | | **US** | **HBMWQ** | **FHH** | | | |
| Intercept | −11.177 | 11.335 | −0.986 | 0.331 | 8.500 | −1.315 | 0.197 | ------ |
| FHH | 1.225 | 0.349 | 3.510 | 0.001* | 0.303 | 4.040 | <0.001* | 2.297 |
| HBMWQ | 0.909 | 0.142 | 6.417 | <0.001* | 0.099 | 9.158 | <0.001* | 2.297 |
| | | | **UH** | **FHH** | **HBMWQ** | | | |
| Intercept | 53.887 | 14.038 | 3.838 | 0.001* | 10.669 | 5.051 | <0.001* | ----- |
| FHH | 0.408 | 0.432 | 0.945 | 0.352 | 0.338 | 1.209 | 0.235 | 2.297 |
| HBMWQ | 0.501 | 0.175 | 2.854 | 0.007* | 0.122 | 4.096 | <0.001* | 2.297 |

UW unimproved water; US unimproved sanitation; UH unimproved hygiene; HBMWQ households below middle wealth quintile; FHH female-headed households; HRR households with rural residence; HWLLE households with low level of education

**Table 4. Selected model parameters for the OLS and MGWR, showing models' goodness of fit.**

| Criterion | OLS | | | | MGWR | | | |
|---|---|---|---|---|---|---|---|---|
| | UW HBWQ-HRR | UW HLLE-HRR | US HBWQ-FHH | UH FHH-HBWQ | UW HBWQ-HRR | UW HLLE-HRR | US HBWQ-FHH | UH FHH-HBWQ |
| Adj. R² | 0.599 | 0.559 | 0.545 | 0.208 | 0.762 | 0.678 | 0.604 | 0.184 |
| AICc | 78.306 | 81.814 | 82.998 | 103.511 | 68.312 | 76.993 | 79.359 | 103.594 |

Note: UW unimproved water; US unimproved sanitation; UH unimproved hygiene; AIC Akaike Information Criterion; HBMWQ households below middle wealth quintile; FHH female-headed households; HRR households with rural residence; HWLLE households with low level of education

of all the predictive models could only account for 76% of the regression plane. Meaning that, there are other factors influencing WASH conditions in Nigeria.

## Descriptive statistics and spatial pattern of WASH conditions and the examined household factors

Fig 2 and Table 5 present (in percentage) the prevalence of the examined individual household factors among the sampled households in Nigeria. According to results, HBMWQ (CV = 71.97), HWLLE (CV = 71.69), UW (CV = 60.30), and FHH (CV = 56.15) were highly heterogeneous. Whereas, HW6M (CV = 9.42), UH (CV = 25.69), HHA60Y (CV = 26.26), HWRR (CV = 45.81) and US (CV = 48.24) were less heterogeneous over Nigeria. Cartogram outputs (Fig 3) show that the highest percentages of households that were getting water from unimproved sources are found in Northern Nigeria. Generally, the percentages of households that were using unimproved water were far lower in the southwest and the FCT. The percentages of households that were using unimproved sanitation are generally high in most of the northern states (Fig 4). However, unimproved sanitation was equally high in Cross River and Bayelsa, in the south. Whereas, the percentage of households with unimproved sanitation was relatively low across the southwestern states and the FCT. The percentages of households with unimproved hygiene are generally high across all the states, with the highest percentages recorded

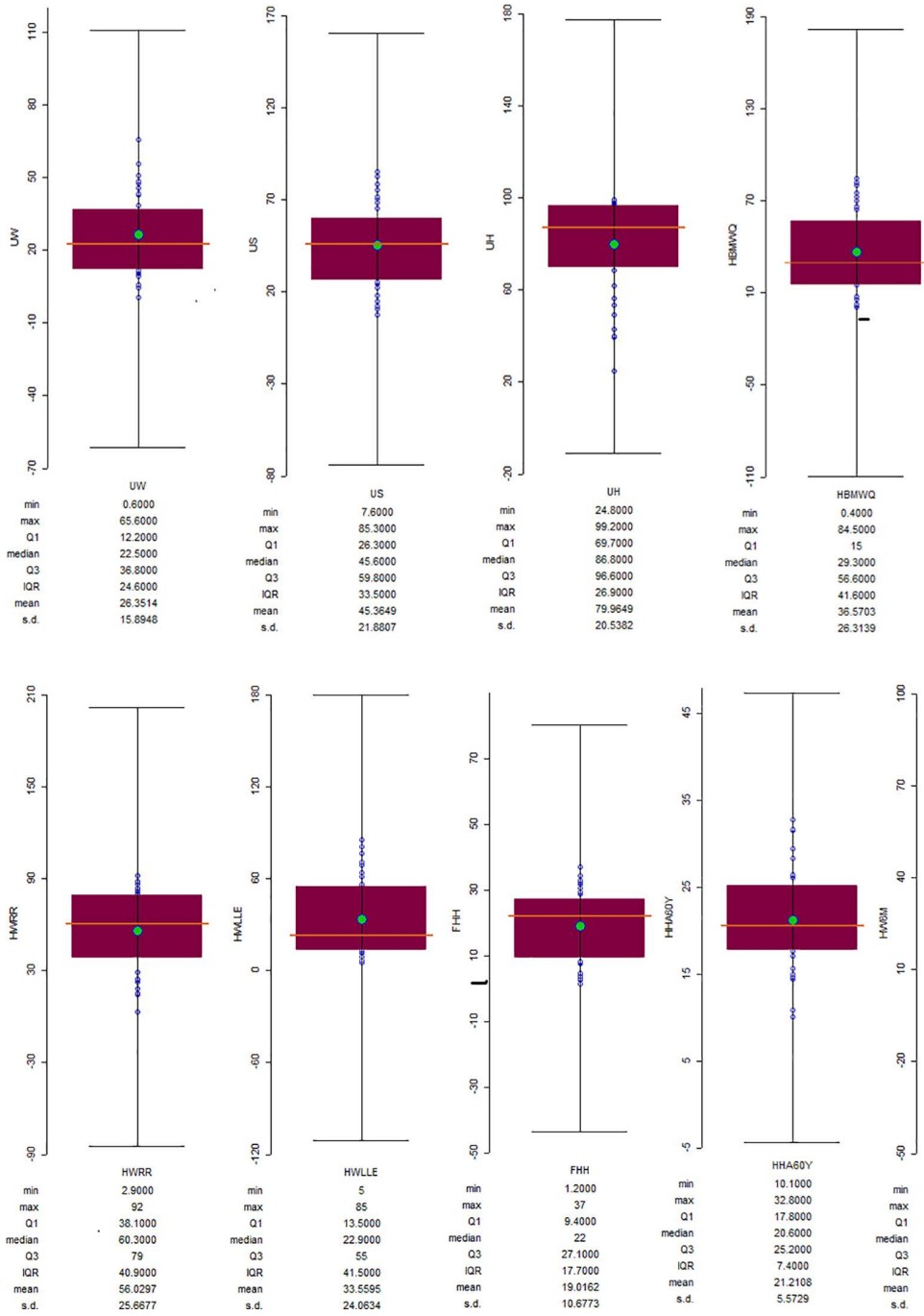

**Fig 2. a. Statistical description of nationwide household factors and WASH conditions.** b. Statistical description of nationwide household factors and WASH conditions.

for Sokoto, Zamfara, Jigawa, Kano, Adamawa, Taraba, Benue, Cross River and Ondo, in the southwest. However, unimproved hygiene was relatively low in Osun, Akwa-Ibom and FCT. The percentages of HBMWQ were highest in the northwest (i.e., Sokoto, Zamfara, Katsina, Kebbi and Niger) and north-east (Borno, Yobe, Gombe and Bauchi). In a sharp

**Table 5. Descriptive statistics of WASH conditions.**

|      | UW    | US    | UH    | HBMWQ | HWRR  | HWLLE | FHH   | HHA60Y | HW6M  |
|------|-------|-------|-------|-------|-------|-------|-------|--------|-------|
| Min  | 0.60  | 7.60  | 24.80 | 0.40  | 2.90  | 5.00  | 1.20  | 10.10  | 5.30  |
| Max  | 65.60 | 85.30 | 99.20 | 84.50 | 92.00 | 85.00 | 37.00 | 32.80  | 43.60 |
| Mean | 26.35 | **45.36** | **79.96** | **36.57** | **56.03** | **33.56** | 19.02 | 21.21  | 21.87 |
| SD   | 15.89 | 21.88 | 20.54 | 26.31 | 25.67 | 24.06 | 10.68 | 5.57   | 2.06  |
| CV   | **60.30** | 48.24 | 25.69 | **71.94** | 45.81 | **71.69** | **56.15** | 26.26  | 9.42  |

UW unimproved water; US unimproved sanitation; UH unimproved hygiene

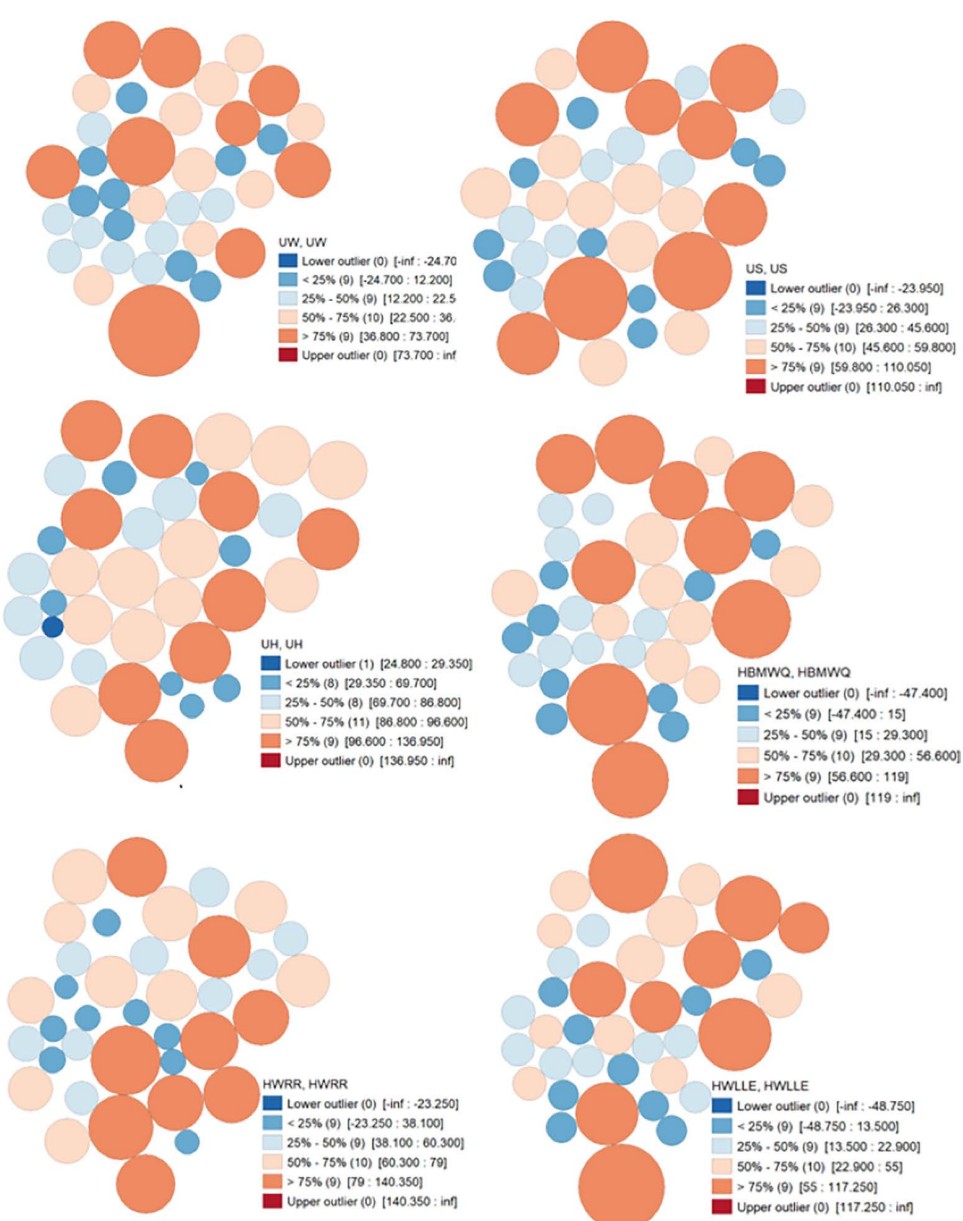

**Fig 3. Thematic cartograms of the spatial quantitative description of household factors and WASH conditions.**

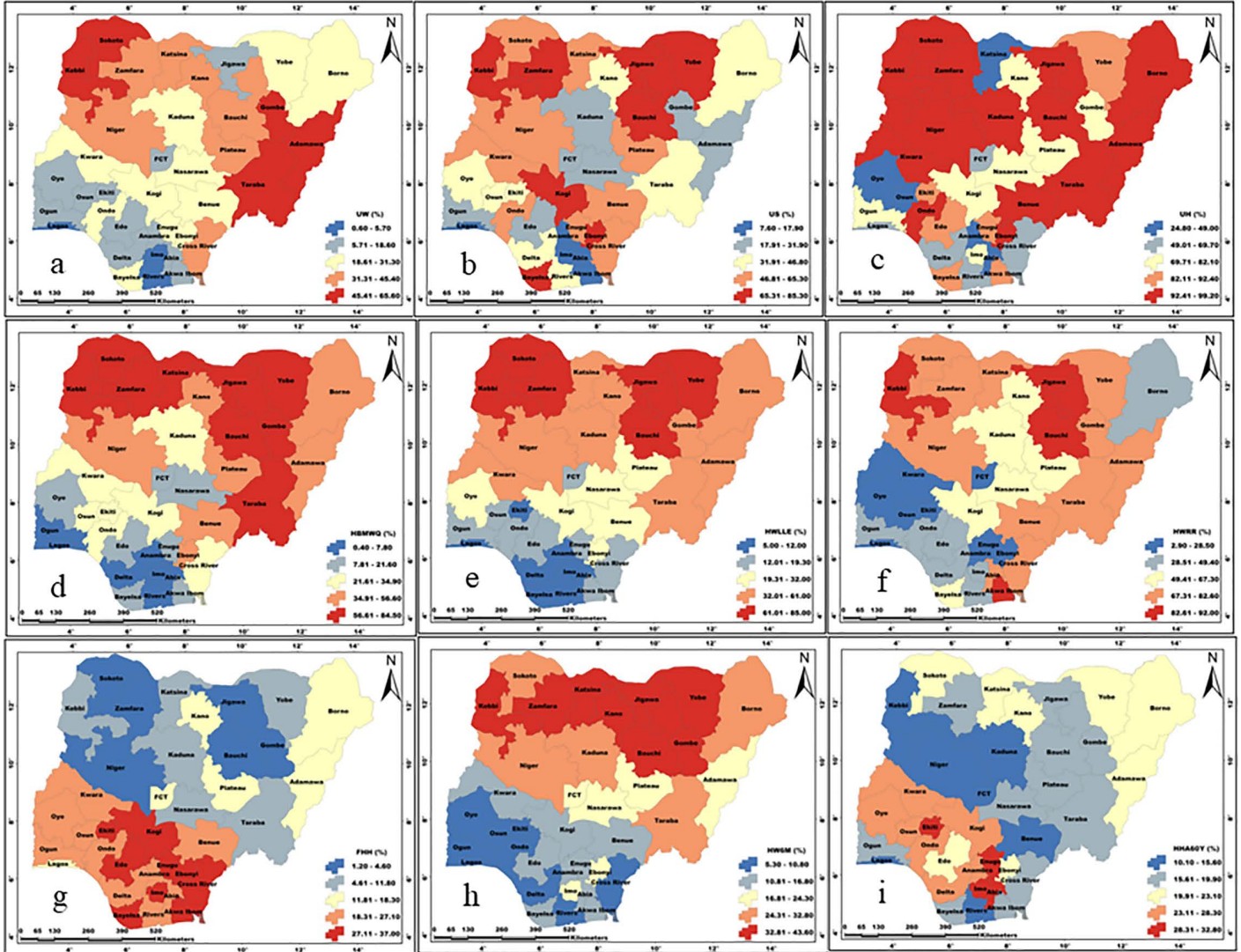

**Fig 4. Spatial pattern of a) UW, b) US, c) UH, d) HBMWQ, e) HWLLE, f) HWRR, g) FHH, h) HW6M and i) HHA60Y.** These maps were created on the ArcGIS 10.8 platform.

contrast, the percentage of HBMWQ was relatively low in the south and the FCT. The percentages of households with rural residence are higher in Sokoto, Kebbi, Yobe, Borno and Gombe, in the north, as well as Benue, Ebonyi, Cross River, Abia and Akwa-Ibom, in the south. In contrast, the lowest percentages of households with rural residence are recorded for the south-western states, Enugu, Imo and Anambra, in the south-east, as well as the FCT. The percentages of households with low level of education (HWLLE) are conspicuously higher in the north-west (Sokoto, Kebbi, Zamfara and Niger) and north-east (Jigawa, Yobe, Borno, Gombe and Bauchi). On the other hand, HWLLE is relatively low across the southern states, except for Ebonyi, which is characterised by relatively higher HWLLE. The results of box plot and cartogram analyses indicate that all the examined parameters (i.e., WASH conditions and household factors) are characterised by both upper (positive) and lower (negative) outliers, emphasising that WASH conditions and household factors significantly varied across states, even within the same geo-political zone in Nigeria. Statistical results reveal that HW6M, FHH and UH are less heterogeneous and characterised by outliers that are relatively closer to the mean across the states of Nigeria.

***Correlations among the household factors and WASH conditions:*** The correlation analysis results are presented in Table 6 and Fig 5. Statistics show that UW exhibited a positive and significant relationship with HBMWQ (0.76 at α = 0.000), HWLLE (0.71 at α = 0.000), HWRR (0.63 at α = 0.000) and HW6M (0.61 at α = 0.000). US exhibited a significant and positive relationship with HBMWQ (0.64 at α = 0.000) and HWLLE (0.52 at α = 0.001). UH exhibited a positive and significant association with HBMWQ (0.48 at α = 0.003) and HWLLE (0.46 at α = 0.004). In addition to the WASH conditions, HBMWQ also exhibited a positive and significant association with HWLLE (0.92 at α = 0.000), HW6M (0.85 at α = 0.000) and HWRR (0.62 at α = 0.000). Aside from the WASH conditions and HBMWQ, HWLLE also exhibited a positive and significant association with HW6M (0.87 at α = 0.000) and HWRR (0.54 at α = 0.000). In addition to the direct association with

**Table 6. Correlations among household factors and WASH conditions in Nigeria.**

|  | UW | US | UH | HHA60Y | FHH | HBMWQ | HWLLE | HW6M |
|---|---|---|---|---|---|---|---|---|
| US | 0.435** |  |  |  |  |  |  |  |
| UH | 0.403* | 0.395* |  |  |  |  |  |  |
| HHA60Y | −0.234 | −0.086 | −0.184 |  |  |  |  |  |
| FHH | **−0.542\*\*** | −0.223 | −0.270 | **0.511\*\*** |  |  |  |  |
| HBMWQ | **0.762\*\*\*** | **0.644\*\*\*** | **0.482\*\*** | −0.278 | **−0.751\*\*\*** |  |  |  |
| HWLLE | **0.706\*\*\*** | **0.524\*\*** | **0.457\*\*** | −0.252 | **−0.810\*\*\*** | **0.916\*\*\*** |  |  |
| HW6M | **0.608\*\*\*** | 0.391* | 0.413* | −0.306 | **−0.824\*\*\*** | **0.848\*\*\*** | **0.870\*\*\*** |  |
| HWRR | **0.628\*\*\*** | 0.277 | 0.335* | −0.357* | **−0.524\*\*** | **0.616\*\*\*** | **0.541\*\*** | **0.550\*\*\*** |

UW unimproved water; US unimproved sanitation; UH unimproved hygiene

*Cell Contents: Pearson correlation,*

*\* p < 0.05; \*\* p < 0.01; \*\*\* p < 0.001*

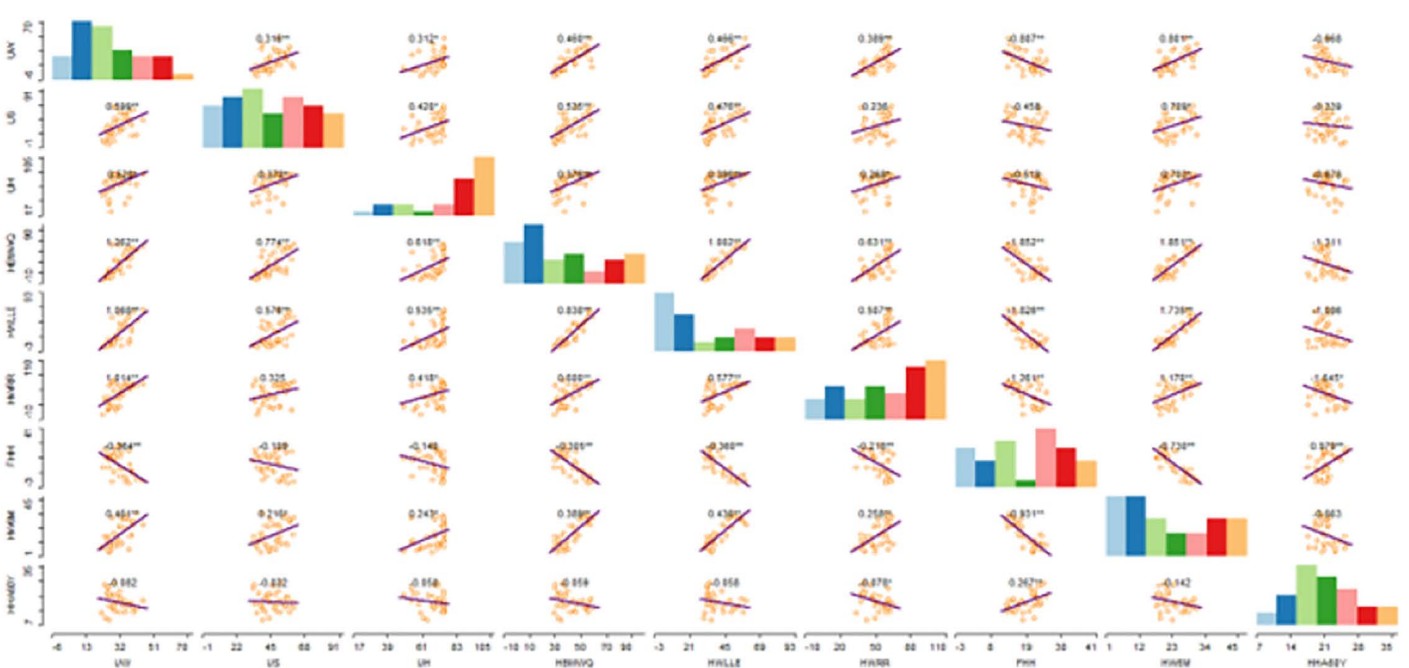

**Fig 5. Cross-plot matrix of household factors and WASH conditions in Nigeria.**

WASH conditions, HBMWQ and HWLLE, HWRR also exhibited a positive and significant association with HW6M (0.55 at α = 0.000). In contrast, FHH exhibited negative but significant association with UW (- 0.54 at α = 0.001), HBMWQ (- 0.75 at α = 0.000), HWLLE (- 0.81 at α = 0.000), HW6M (- 0.82 at α = 0.000), and HWRR (- 0.52 at α = 0.001). However, FHH exhibited a positive and significant association with HHA60Y (0.51 at α = 0.001).

### The spatial pattern of model performance

The spatial patterns of local $R^2$ values that are recorded for all the regression models are presented in Figs 6 - 8. The performances of the regression models obviously vary spatially, and the magnitude of performance differs from one model to the other. Thus, the model results are better presented based on individual WASH conditions.

**Model performance: Unimproved water.** For UW, we came up with two models – UW ≈ HBMWQ.HWRR and UW ≈ HWLLE.HWRR (Fig 6). For the first model, local $R^2$ values range from 0.60 to 0.90, with the highest values occurring in Sokoto, Kebbi, Zamfara, Niger, Kwara and Oyo states and the least values are recorded for Borno, Yobe, Gombe, Adamawa, Bayelsa, Rivers and Akwa-Ibom. Whereas a range of 0.42–0.88 local $R^2$ values is computed for the second model, with a similar spatial pattern to that of the first model. However, results show that HBMWQ and HWRR exhibit stronger causal relationships with UH compared to HWLLE and HWRR.

**Model performance: Unimproved sanitation.** A range of 0.58–0.70 local $R^2$ values is recorded for the model US ≈ HBMWQ.FHH, with a spatial pattern that is similar to that of UW spatial predictive models, however, with a slight difference (Fig 7). First, there is an obvious West-East oriented increase in model performance. And secondly, the lowest $R^2$ values are restricted to Borno and Adamawa states in the north-east.

**Model performance: Unimproved hygiene.** A range of 0.250–0.253 local $R^2$ values is recorded for the model UH ≈ FHH.HBMWQ, which is quite low (Fig 8). Here, the spatial pattern of model performance is completely different from that of other models. The local $R^2$ is characterised by a close range that indicates relative homogeneity. However, UH spatial regression model is characterised by South-North directional performance, with the higher local $R^2$ values occurring in Delta, Bayelsa, Rivers, Akwa-Ibom, Abia and Imo in the south; while the lowest are recorded for Sokoto, Zamfara, Katsina, Kano, Jigawa, Yobe and Borno in the north.

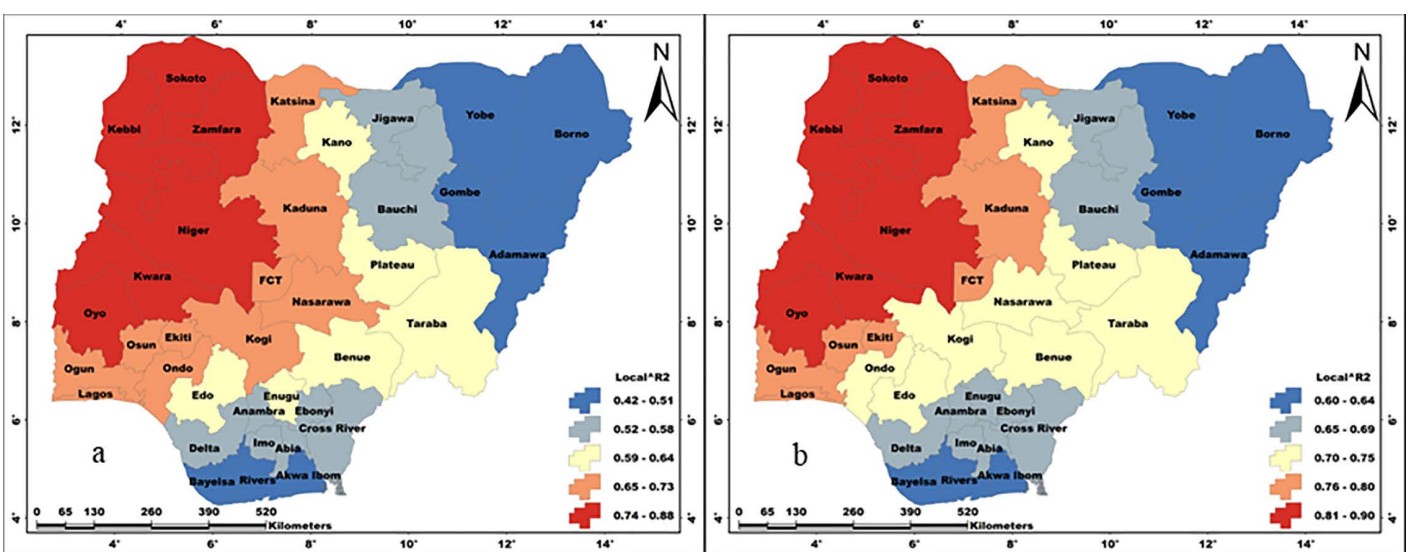

**Fig 6. The relationships between unimproved water and a) HBMWQ-HWRR and b) HWLLE-HWRR.** These maps were created on the ArcGIS 10.8 platform.

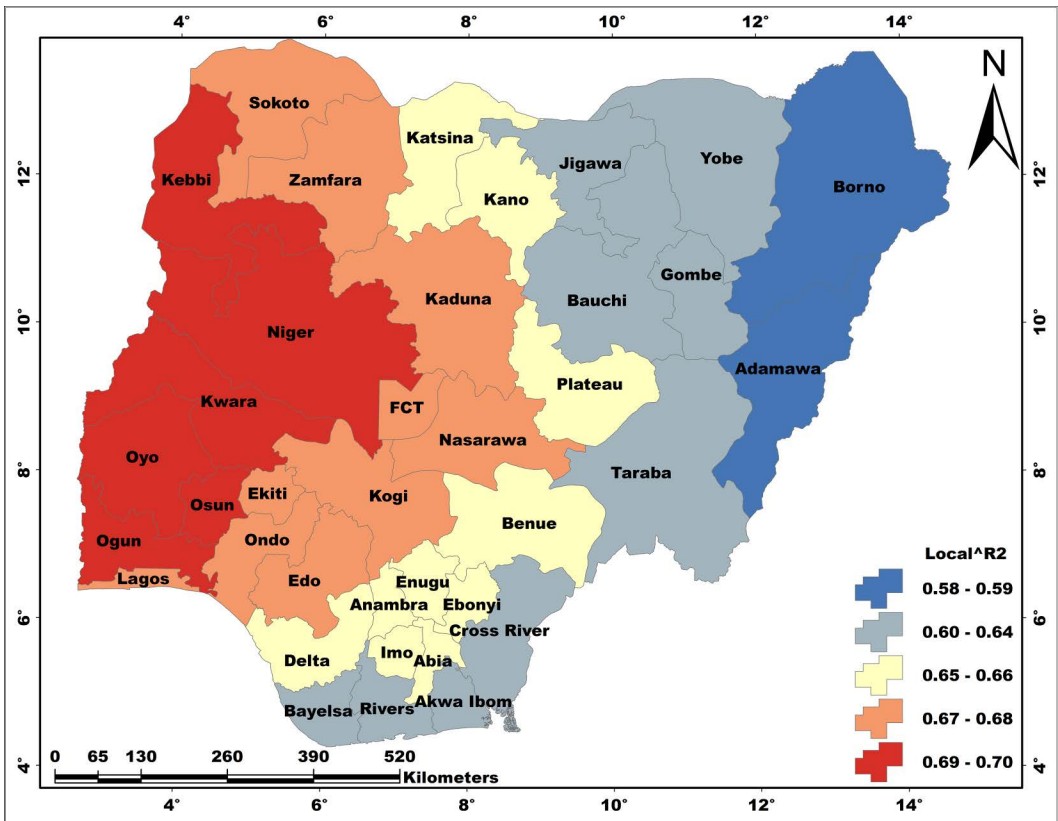

**Fig 7. The relationship between unimproved sanitation and WQHH.** This map was created on the ArcGIS 10.8 platform.

## Influence of household factors on WASH conditions

The OLS statistical indices that show the P-values, Adjusted $R^2$ and AICc of the causal relationships between household factors and WASH conditions are presented in Table 7. The statistics show that HBMWQ, HWLLE and HWRR exhibit a significant causal relationship with UW at $\alpha = 0.000$. In the same vein, HBMWQ and FHH exhibit significant causal relationships with the US at $\alpha = 0.000$. Likewise, HBMWQ and FHH exhibit causal relationships with UH at $\alpha = 0.004$ and 0.345, respectively. The computed Adjusted $R^2$ and AICc values range from 0.599 to 0.208 and 78.31 to 103.51, respectively.

The computed MGWR's degree of dependence (DoD_j) of the predictor variables is presented in Table 8. The parameter degree of dependence shows that UW mostly depends on HWRR, with a DoD_j value of 0.998. Whereas, the degrees of dependence of UW on HWLLE and HBMWQ are 0.535 and 0.485, respectively. Whereas, the computed DoD_j (0.995 and 0.996) reveal that US and UH solely depend on HBMWQ. The models' DOD values range from 0.714 to 0.996, with Adjusted $R^2$ and AICc value ranges of 0.762–0.184 and 68.312–103.594, respectively.

The coefficient maps of the influencing household factors are presented in Figs 9-11. The maps present the spatial patterns of the individual influence of the examined household factors on the WASH conditions in Nigeria. We analysed the spatial pattern of parameter influence on the basis of individual WASH conditions.

**Unimproved water.** The influence of HBMWQ on UW is highest in southwestern states (Oyo, Ekiti, Osun, Ogun and Lagos) and central-southern states (Imo, Abia, Rivers and Akwa-Ibom). Results show that HBMWQ exerted very high and significant influence on UW in southern and central parts of Nigeria (Fig 9). On the other hand, the influence of HBMWQ

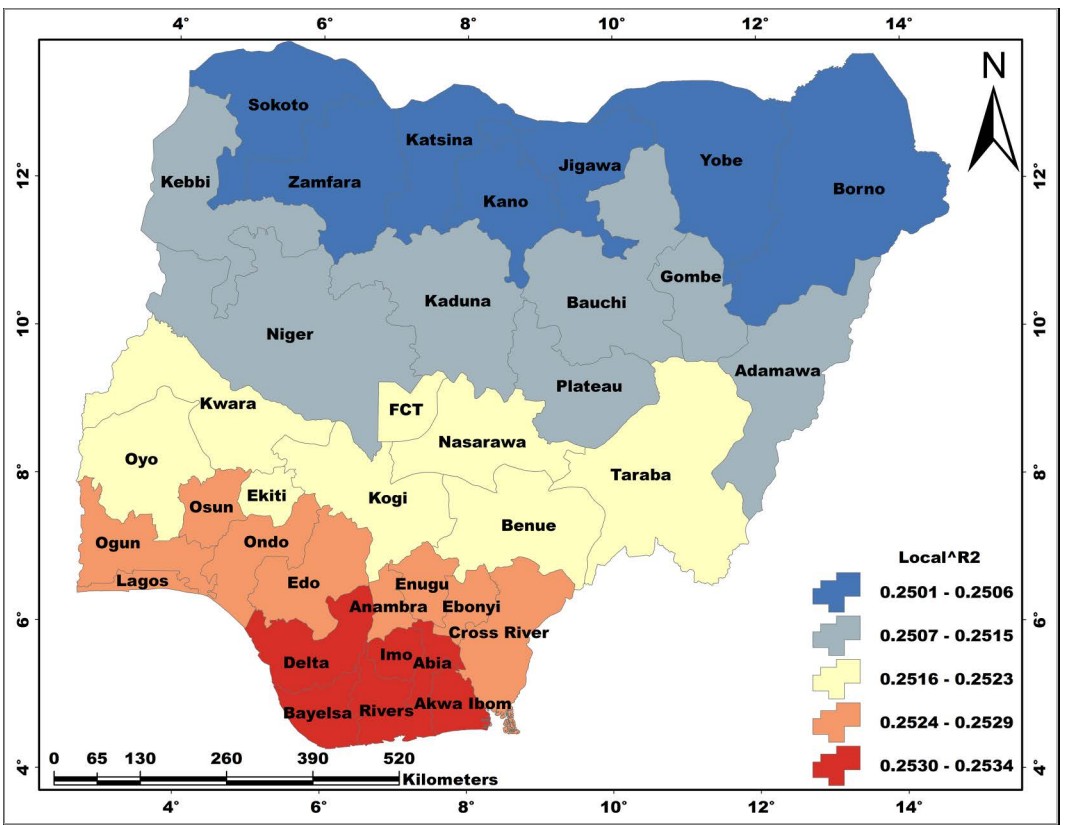

**Fig 8. The relationship between unimproved hygiene and FHHWQ. This map was created on the ArcGIS 10.8 platform.**

**Table 7. Household factors' influence on WASH conditions in Nigeria.**

| Variables | P-value | | |
|---|---|---|---|
| | UW | US | UH |
| Intercept | 1.000 | 1.000 | 1.000 |
| HBMWQ/HWLLE | 0.000 | 0.000 | 0.004 |
| HWRR | 0.000 | – | – |
| FHH | – | 0.000 | 0.345 |
| AICc | 78.306 | 82.998 | 103.511 |
| Adj. R$^2$ | 0.599/0.559 | 0.545 | 0.208 |

Note: AIC – Akaike Information Criterion, Adj. R2 - Adjusted R2 as the coefficient of determination

on UW is very low and insignificant in the north. Generally, the influence of HBMWQ on UW has a SW-NE oriented spatial pattern. The spatial pattern of HWLLE's influence on UW is similar to that of HBMWQ, but with a slight difference. For HWLLE, parameter influence is higher across most of the south-eastern and south-southern states (i.e., Enugu, Anambra, Ebonyi, Imo, Abia, Akwa-Ibom, Cross River, Rivers and Bayelsa), as well as in Ogun and Lagos states in the south-west. The influence of HWRR on UW is generally low and less heterogeneous across Nigeria. Nevertheless, the highest influence of HWRR is delineated in the south, across Rivers, Bayelsa, Delta, Akwa-Ibom, Imo, Abia, Cross River

**Table 8. WASH conditions' degree of dependence on household factors in Nigeria.**

| | DoD_j | | |
|---|---|---|---|
| Variables | UW | US | UH |
| Intercept | 0.998 | 0.737 | 0.997 |
| HBMWQ | 0.485 | 0.996 | 0.995 |
| HWLLE | 0.535 | – | – |
| HWRR | 0.998 | – | – |
| FHH | – | 0.997 | 0.997 |
| DoD | 0.714 | 0.881 | 0.996 |
| AICc | 68.312 | 79.359 | 103.594 |
| Adj. R² | 0.762 | 0.604 | 0.184 |

Note: AIC – Akaike Information Criterion; DoD_j – degree of dependence; Adj. R2 – Adjusted R2 as the coefficient of determination.

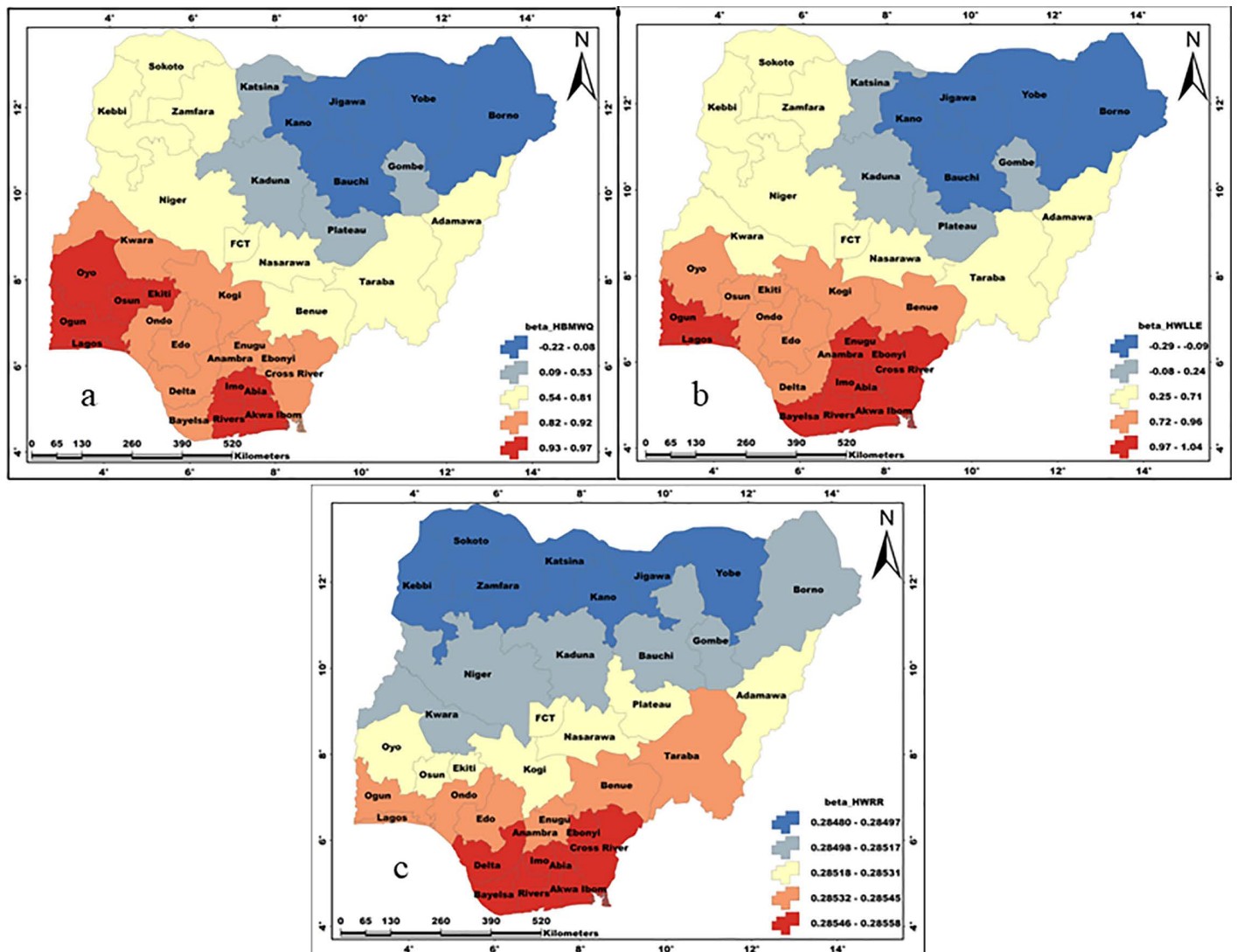

**Fig 9. Influence of a)HBMQ, b)HWLLE, and c) HWRR on unimproved water.** These maps were created on the ArcGIS 10.8 platform.

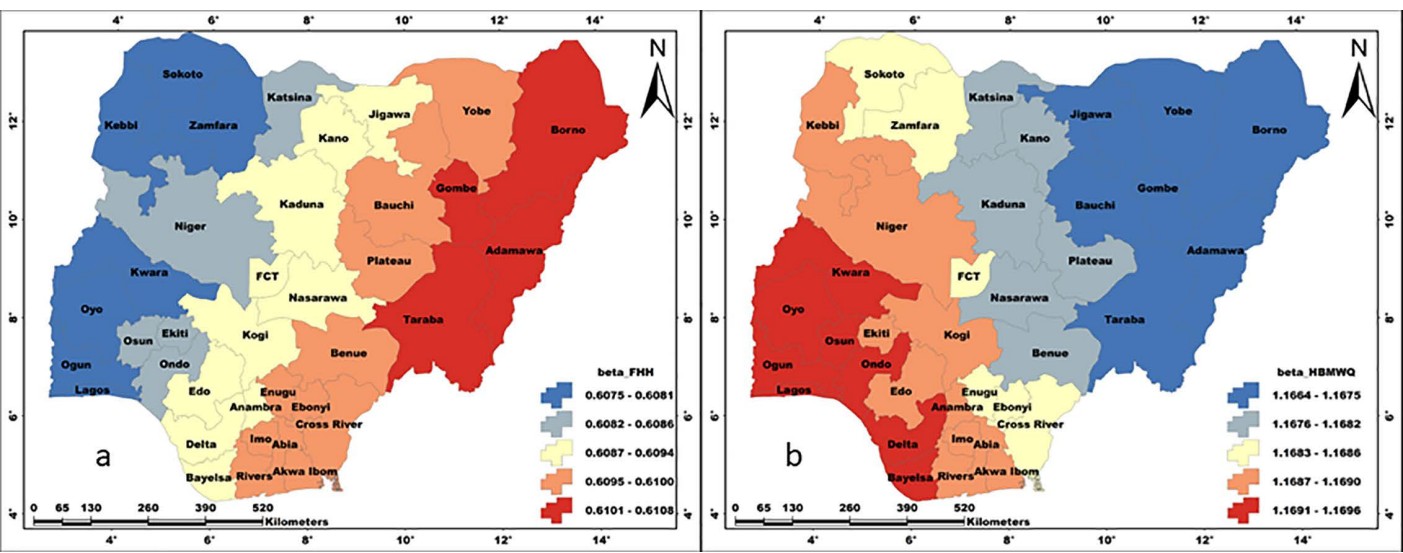

**Fig 10. Influence of a) HBMWQ and b) FHH on unimproved sanitation.** These maps were created on the ArcGIS 10.8 platform.

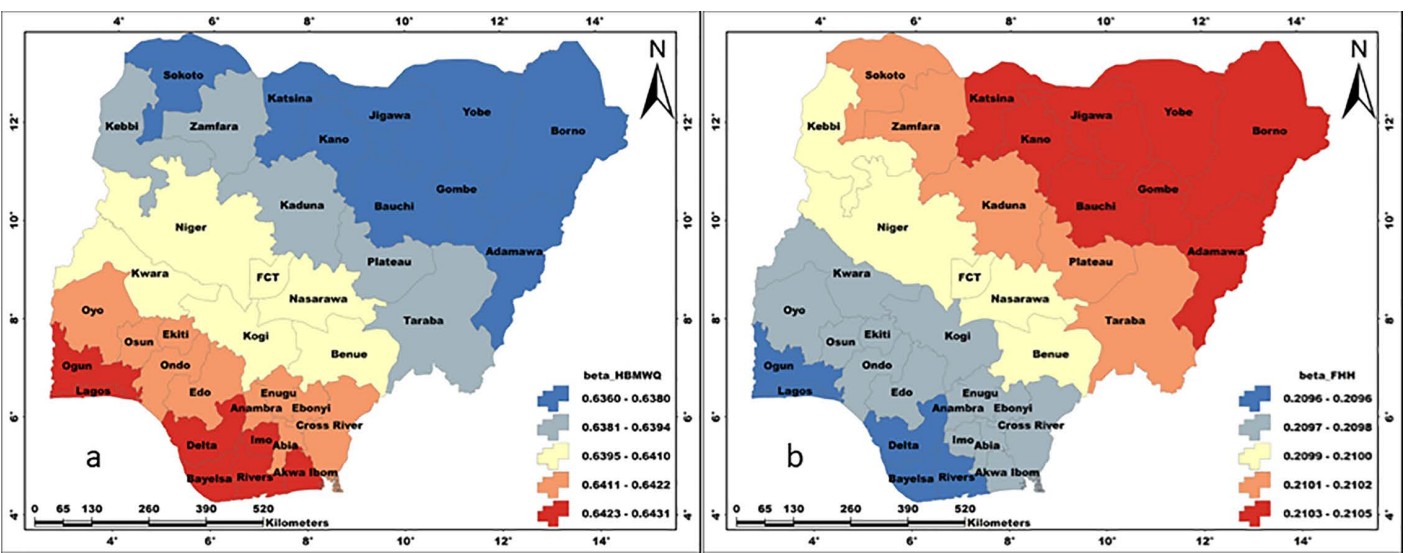

**Fig 11. Influence of a) HBMWQ and b) FHH on unimproved hygiene.** These maps were created on the ArcGIS 10.8 platform.

and Ebonyi states. Whereas, the least influence of HWRR on UW is common to the northern states of Sokoto, Katsina, Jigawa, Yobe, Kebbi, Zamfara and Kano.

**Unimproved sanitation.** The influence of HBMWQ on US is generally high and less heterogeneous, exhibiting a W-E spatial pattern with highest and the lowest coefficients occurring in the south-west (Oyo, Ogun, Ondo, Lagos and Kwara) and north-east (Borno, Adamawa, Taraba, Gombe, Bauchi, Yobe and Jigawa) respectively (Fig 10). The influence of FHH on US is significantly high but less heterogeneous. In direct contrast to the influence of HBMWQ on US, the influence of FHH on US exhibits an E-W spatial pattern of occurrence. In this case, FHH exerts higher influence on US in the

north-eastern states (i.e., Borno, Adamawa, Taraba and Gombe). Whereas the least influence occurs in north-west (Sokoto, Kebbi and Zamfara) and south-west (i.e., Lagos, Ogun, Oyo and Kwara).

**Unimproved hygiene.** The coefficient of HBMWQ on UH is generally high across Nigeria. Hence, HBMWQ's influence on UH is less heterogeneous but with a SW-NE spatial pattern of occurrence. HBMWQ's coefficient is specifically higher in Delta, Bayelsa, Rivers, Akwa-Ibom and Imo states in the south. Whereas, the least coefficient is observed in Borno, Adamawa, Yobe, Gombe, and Bauchi states in the north-east as well as Katsina, Sokoto and Kano states in the north-west (Fig 11). The coefficient of FHH on UH is generally low and less heterogeneous over Nigeria. In direct contrast to HBMWQ's coefficient on UH, the spatial pattern of FHH's coefficient on UH has a NE-SW orientation, with higher influence in north-east (Borno, Adamawa, Yobe, Gombe and Bauchi) as well as in Kano and Katsina states. Whereas, the least of the influence of FHH on UH is found in South-South (i.e., Delta, Bayelsa and Rivers) and South-Western states of Ogun and Lagos.

## Discussion

WASH conditions in Nigeria differ by space and depend not only on resource availability but also on household mindfulness, affordability and sustainability. This study investigates the relationship between household factors and WASH conditions. It also evaluates the spatial context and models the spatial pattern of both in Nigeria. Understanding the spatial patterns and the contributions of the household factors will serve as a policy guide for addressing the gaps in intervention programmes in Nigeria.

The available data indicate low literacy and high poverty levels despite the claim that Nigeria has the biggest economy in Africa. However, the country still has the potential for development if it can improve its education and economic growth. Despite the increasing rural-urban migration, the majority of Nigerians still reside in rural areas, and many of the so-called cities and towns in the country lack the required standards. In this case, many cities and towns lack basic amenities, including potable water supply systems and good road networks. Analyses show that there were more older adults in North-West, South-East and South-West, while the lowest percentage of older adults was recorded for North East (10%) and South-South. The distribution of older adults has a similar spatial pattern to that of the crime rate and social restiveness in Nigeria. For instance, the North East has been the epicentre of insurgency and banditry for more than a decade, and the most vulnerable people are elderly persons who are physically weak as a result of old age. Whereas in the South-South, the ongoing environmental degradation as a result of oil pollution and the consequent deterioration in public health may have resulted in low life expectancy in the oil-producing areas.

Our study shows that, generally, while 2 in 3 households were using improved water sources, over half of the households were doing the same for sanitation in Nigeria. Environmental and personal hygiene conditions are still very poor, as only 23% of the sampled households were practising improved hygiene. This indicates that unimproved hygiene is rampant among the people across all socioeconomic strata in Nigeria. Our data agree with the findings of Rahut and colleagues [41] who noted a more significant progress in access to basic drinking water compared with other WASH facilities across poor countries in Africa and Asia. It also aligns with Olaitan and colleagues [42], who observed that sanitation practice was limited in Nigeria. However, there were some spatial variations between northern and southern Nigeria. Despite the general improvement in the use of improved water sources, our findings show that more than half of the population in some northern states, including Adamawa, Bauchi, Gombe, Sokoto and Taraba, still depend on unimproved water sources. The pattern is similar for unimproved sanitation, which is a common place in northern states such as Bauchi, Kogi, Kebbi, Jigawa, Yobe and Zamfara, except two states in the south, Bayelsa and Ebonyi States. The spatial patterns may be linked to the fact that two-thirds or more of the households in the affected states were rural residents, except Ebonyi State (22%), as shown in the supplementary Table 1. Unhygienic practices, although generally high across Nigeria, follow a similar pattern with the highest prevalence in the north and a few southern states.

This study shows that WASH conditions are generally influenced by HBMWQ, HWLLE, HWRR and FHH, indicating the roles of household financial status, level of education, living conditions and the gender dynamics in WASH conditions in Nigeria. Specifically, the quality of domestic water is a function of economic and educational status as well as the residence type of the household, whereas the choice of sanitation and hygiene practice largely depends on economic status and the gender of the household head. The findings implied that the use of unimproved water, sanitation and hygiene will increase with an increase in the number of households living below the poverty level and with low levels of education. Thus, WASH conditions are predominantly determined by the economic status of households in Nigeria. In line with our result, Wada et al. [43] had previously observed that inequality in access to WASH in Nigeria is driven by socioeconomic inequalities. The outcome of our analyses clearly shows that unimproved hygiene is less associated with economic and educational attainment in Nigeria. This finding suggests that efforts to improve hygienic practices should extend beyond the present interventions. In this case, environmental and health education and awareness campaigns must be a focus of subsequent interventions.

Our study also shows that causal relationships between household factors and WASH conditions vary significantly from one state to the other in Nigeria. This is proof that spatial interaction is location-specific and should be analysed as an independent relationship that varies in space and time. Thus, a relationship of this nature should be examined with predictive models that consider location as an inevitable clause in the modelling of spatial relationships [40]. Our adopted local regression model (i.e., MGWR) indicates a better performance than its global counterpart (i.e., OLS), corroborating the fact that the relationships between household factors and WASH conditions vary significantly and is location location-specific. Both the local and global regression models jointly emphasised the existence of causal relationships between household factors and WASH conditions in Nigeria.

The highest model performance is attained by the UW $\approx$ HBMWQ.HWRR with a minimum of 60% and a maximum of 90% explanation of the regression plane. Altogether, the UW predictive models affirm that accessibility to potable water is determined by economic status, literacy level and urbanisation. This is because job and business opportunities are mostly limited to urban areas where the literacy level is relatively high compared to rural areas in Nigeria. This indicates that policy efforts and development opportunities largely concentrate in urban centres. Consequently, poor rural dwellers with limited social amenities predominantly rely on water from unimproved sources in Nigeria. Poignantly, this is true as more than 90% of rural dwellers do not have access to potable water supply in the country [10,15,16]. Results show that HBMWQ is the only household factor that has a significant influence on all the WASH conditions. This implies that WASH conditions are determined by the economic status of the people. The computed degree of dependence (DoD = 1) shows that UW is predominantly dependent on HWRR (see Table 2). This implies that the use of unimproved water is higher among rural households than their urban counterparts. On the other hand, US and UH are dependent on HBMWQ (with DoD = 1), meaning that both are predominantly linked to poverty in Nigeria. Similarly, the choice of sanitation and hygiene is highly dependent on the gender of the household head. The computed statistical indices (i.e., Adjusted $R^2$ and AICc) show that the local regression models have higher prediction strength and greater model reliability. Based on the higher Adjusted $R^2$ computed for the OLS-based UH predictive model, the influence of HBMWQ and FHH on UH is not absolutely location-specific in Nigeria. Thus, the observed poor performance of UH predictive models suggests the need for stakeholders to look beyond household factors in tackling the challenge of unimproved hygiene.

Geospatial modelling shows that the predictive models for UW recorded the highest performances in the north-west and south-west, while their lowest performances occurred in the north-east and south-south geo-political zones. The observed spatial pattern of model performances in the north-east could be attributed to terrorism and banditry that trigger the ongoing forced migration of people to IDP camps, where accessibility to safe water, sanitation and hygiene is extremely low. In contrast, the recorded low model performances in the south-south could be attributed to the environmental conditions and the homogeneity of people's attitude towards WASH regardless of socioeconomic status, especially in the Niger Delta. The predictive model for US also showcases high performance, however, with a shorter range and a

lower upper band of $R^2$ values compared to that of UW. The interpretation is that the explanatory variables exhibit strong causal relationships with the US across all Nigeria's states. Like UW predictive models, US predictive model performance exhibits a west-east oriented spatial pattern, however, with a slight difference. While the highest performances occurred in the west, the lowest performances are restricted to the north-east. In this case, model performance decreases from the west to the east. The observed poor performance of the UH predictive model reveals that poor hygiene cannot be solely blamed on household factors in Nigeria. The extremely low range of the computed $R^2$ values is an indication that the unimproved hygiene is not peculiar to people of certain socioeconomic classes. Rather, hygiene is more of an attitude and societal perception. Nevertheless, the spatial pattern of model performance for UH is completely different from that of UW and US. The highest model performances occur in the south-south, and the lowest performance is found in the Sahel region of both the east and the west of northern Nigeria. In this case, model performance decreases from the south to the north. This implies that UH could partly be predicted by economic status and the gender of household heads in Nigeria.

The spatial patterns of parameter influence on UW reveal that accessibility to potable water largely depends on the economic status and literacy level of households in southern Nigeria. Whereas economic status and literacy level do not have a significant influence on the use of potable water in north-eastern Nigeria. Thus, investing in education and economic empowerment could lead to significant improvement in household WASH conditions in the south. However, there is more to do in addition to investing in education and economic empowerment of the people in the north. Addressing the problem of water scarcity and embarking on WASH education and awareness could be some of such promising investments in northern Nigeria. Our study reveals that the use of unimproved water is common among rural households in the south, particularly in the Niger Delta. Meaning that accessibility to potable water is relatively high in urban centres of southern states, emphasising the developmental dichotomy that exists between the rural and urban areas, particularly in southern Nigeria. In contrast, results show that the use of unimproved water is not peculiar to rural households in northern Nigeria. In this case, the accessibility to improved water has not much to do with residence type in northern Nigeria.

The observed spatial pattern of influence of HBMWQ and FHH on the US is an indication that the choice of sanitation is strongly determined by economic status and the gender of the household head in the Southwest and Northeast, respectively. The coefficient maps show that the influence of HBMWQ on US is generally high across the states of Nigeria. Thus, economic status has an overwhelming influence on the choice of sanitation across all the states of Nigeria. However, the gender of the household head is equally a significant influencing factor on the choice of sanitation. The influence of HBMWQ and FHH on unimproved hygiene presents contrasting spatial patterns. While the highest HBMWQ's influence occurs in the South-west, that of FHH is delineated in the North-east. Just like the case of other WASH conditions, unimproved hygiene is associated with poverty, particularly in southern Nigeria. We observe that economic status influences the WASH conditions in Nigeria, but with a profound magnitude in the south. Although the influence of FHH on UH is generally low and less heterogeneous across the country, it is evident that the relationship between hygiene and the gender of the household head is more pronounced in north-eastern Nigeria. The spatial pattern of individual parameter influences firmly conforms to the form of correlation that exists among the examined parameters. For instance, HBMWQ and HWLLE are strongly correlated, and they both exhibit similar spatial patterns of influence on UW. Similarly, the strong inverse association between HBMWQ and FHH also fall in line with the opposing spatial pattern of parameter influence of the former with the latter.

As indicated by the computed $R^2$ and AICc, the strongest and most reliable predictive model was developed for UW. This means that UW can be reliably predicted by HBMWQ/HWLLE and HWRR in Nigeria. In the same vein, statistical indices affirm that the US predictive model is equally strong and can reliably give a meaningful explanation to sanitation conditions in Nigeria. Regression model results show that the strength of the UH predictive model is low, as it can account for only 25% of the regression plane. This implies that there are other factors contributing to hygiene conditions in Nigeria. Analyses show that HBMWQ is the only household factor that exhibits a strong causal relationship with all the WASH conditions. Interestingly, based on our statistics, HWRR and HWLLE have higher degrees of dependence compared to

HBMWQ. For instance, UW has higher dependence on HWRR and HWLLE, while US and UH also have higher dependence on FHH. This means that despite the dominant influence of HBMWQ on the predictive models, WASH conditions largely depend on HWRR, HWLLE and FHH in Nigeria. In line with our result, Tseole et al [28] observed that the challenge of unimproved WASH is more pronounced in the rural areas of Southern Africa. The inspiration that could be drawn from these findings is that the provision of basic amenities for the rural dwellers, an increase in literacy level and women empowerment could enhance better WASH conditions in low-income countries. These promising interventions will also lift many households above the poverty line.

Existing studies from low- and middle-income countries have examined various aspects of WASH conditions and interventions, including the efficiency of WASH interventions [44–47]; effectiveness of WASH interventions on the health of the beneficiary households [42,43,48,49]; the burden of disease from unimproved WASH in poor countries [27,31,50]; impacts of behavioural change on WASH quality in low-income countries [51]; barriers and facilitators to WASH practices [28,41]; marginalisation of women and girls in WASH policies [26]; and the cost of achieving basic WASH conditions in the least developed countries [52]. However, their emphasis has been on WASH conditions and their influence on the health of the people. Aside from Wada and colleagues [49], who noted the possible influence of household conditions on WASH, we are not aware of an existing study that has examined the influence of household factors on WASH conditions. In our study, we employ a mixture of both spatial and non-spatial inferential statistical models to examine the relationship between household factors and WASH conditions in Nigeria. In addition to the conventional regression indices, we analyse the Degree of Dependence (DoD) to determine the varying influence of household factors on WASH conditions. We also employ a spatial proportional mapping tool (i.e., cartogram) to illustrate the spatial patterns of the examined household factors, WASH conditions and the research outputs. We implement both bivariate and multivariate statistical algorithms to model the dependence of individual WASH conditions on one or more household factors. All the adopted methods affirm that WASH conditions are influenced by household factors in Nigeria. Thus, our study is the first of its kind that employs spatial regression models to explore the variations that exist in the location-specific influence of household factors on WASH conditions.

The reliability of the outcome of research of this kind largely depends on the representativeness of the sample to the population. Although the dependability of our data is confirmed by the agreement of our results with those of previous studies, we realise that comprehensive data, such as demographic and health surveys, will certainly offer more reliable and robust data that can bring the best out of our methods and approach to this study. For instance, the availability of national population and housing census data will offer the opportunity to implement our methods at finer resolutions, such as county and ward levels. In addition, there is an urgent need to examine the disparity in the progressive improvement of individual WASH conditions in the least developed countries of the world. Therefore, we suggest that future studies should unravel the reasons behind the attitude of people towards the practice of personal hygiene in Nigeria and other low-income countries.

In this study, we present an elite analytical method that not only establishes a causal relationship between household factors and WASH conditions but also models the spatial variability of the causal relationship as well as the spatial pattern of individual household parameter influence on WASH conditions. In addition to the conventional statistical indices, we employ DoD to further reveal the individual influence of household factors on the WASH conditions. The adoption of multiple geospatial analytical platforms gives us the opportunity to implement multiple algorithms to explore the internal structure of our data. This novel approach provides us with both numerical and graphical understanding of the spatial pattern of the influence of individual household factors on WASH conditions. The benefit of this approach is that stakeholders are offered a new opportunity to identify region-specific household conditions that could be addressed to improve the WASH conditions. Unlike the non-spatial global regression models, the adopted spatial local regression model was able to reveal the strongest predictor variable as well as the most influential factor for WASH conditions in Nigeria. Nevertheless, the best of the predictive models only accounts for 76% of the variance, indicating that there are other variables contributing

to WASH conditions in Nigeria. Such variables may include factors that influence physical water availability, economic water availability, government policies, and intervention programmes.

### Limitations of the study

This study is based on survey data, although it could have benefited more from a robust analysis with census data. However, due to the unavailability of recent census data in Nigeria, the 2018 DHS remain the most recent and nationally representative dataset available to achieve the same purpose as the census data. Also, the household parameters considered for this study were limited to available variables in the dataset. Additional household variables, including major household occupation, political affiliation, social network and access to government support, are recommended in subsequent studies. In addition to these factors, community factors and state-level policies should also be considered in subsequent studies.

### Implications for improved WASH conditions in Nigeria

This study clearly shows that WASH conditions vary significantly across geopolitical zones and states in Nigeria and are influenced by highly heterogeneous household parameters. Thus, WASH interventions should be community-specific. In addition, the choice and approach to a specific WASH intervention should be based on reliable data. For instance, communities with relatively similar WASH conditions, household settings, socioeconomic status and cultural beliefs would better benefit from similar but suitable WASH interventions. Therefore, it will be logical to set varying time-lines and targets for WASH interventions based on the current WASH conditions, prevailing circumstances and the terms and conditions of the available interventions.

### Implications for household poverty and standard of living in Nigeria

Based on our findings that the majority of Nigerians reside in rural areas with deplorable living and poor WASH conditions, unlike the urban setting, there is an urgent need for bridging the rural-urban gaps in provision of basic amenities, such as potable water, electricity, good road network, sound education and functioning health facilities. The country has a huge potential for development if the government can review its developmental policies for accelerated and sustainable rural development. Drawing from our findings, WASH interventions without a commensurate investment in people's socioeconomic development may yield little or no results. Therefore, we recommend channelling donors' and governments' investment in the socioeconomic development of the communities where dwellers will maximally benefit from WASH interventions.

## Conclusion

This study successfully demonstrates that WASH conditions research and interventions have a lot to benefit from spatial analyses. The study concludes that high-risk or unimproved WASH is associated with rural residence, which is usually characterised by a low level of education, poverty and large household size. Households with these characteristics may not be physically, socially and economically prepared to provide improved WASH for the members due to possible lack of information about the health consequences, financial incapacity and poverty of the environment in which they live. The study concludes that WASH conditions and their influencing household factors exhibit complex relationships and are highly heterogeneous in Nigeria. Thus, WASH interventions should be people-, community- and region-specific, rather than a One-For-All programme. The study further concludes that the high prevalence of unimproved hygiene, irrespective of the household's wealth status and educational level, suggests the need for proper health and hygiene education through all possible means. To improve household WASH conditions in Nigeria, this study suggests rural development-focused policies for all tiers of government in Nigeria. Government and stakeholders should work together to encourage environmental and health education, particularly in rural areas. Rural socioeconomic interventions, such as agri-business,

adult education, women empowerment, provision of improved crop varieties and infrastructural development should be prioritised in Nigeria.

## Supporting information

**S1 File. Descriptive and bivariate statistics of WASH conditions and household factors.**
(DOCX)

## Acknowledgments

The authors appreciate ICF International for granting permission to use the Demographic and Health Survey datasets of all the selected West African countries. The reviewers are hereby acknowledged for their constructive criticisms, valuable comments and suggestions.

## Author contributions

**Conceptualization:** Jacob W. Mobolaji.

**Data curation:** Jacob W. Mobolaji.

**Formal analysis:** Akinola Shola Akinwumiju.

**Investigation:** Jacob W. Mobolaji.

**Methodology:** Akinola Shola Akinwumiju, Jacob W. Mobolaji.

**Project administration:** Akinola Shola Akinwumiju, Jacob W. Mobolaji.

**Resources:** Akinola Shola Akinwumiju.

**Software:** Akinola Shola Akinwumiju, Jacob W. Mobolaji.

**Supervision:** Akinola Shola Akinwumiju, Jacob W. Mobolaji.

**Validation:** Akinola Shola Akinwumiju, Jacob W. Mobolaji.

**Visualization:** Akinola Shola Akinwumiju.

**Writing – original draft:** Akinola Shola Akinwumiju, Jacob W. Mobolaji.

**Writing – review & editing:** Akinola Shola Akinwumiju, Jacob W. Mobolaji.

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
