## [Decision Letter · Decision Letter 0]

23 Feb 2025

PONE-D-24-54290Geospatial Assessment of Household Water, Sanitation and Hygiene Conditions and Associated Factors in Nigeria: A Causal Relationship ModelPLOS ONE

Dear Dr. Akinwumiju,

Thank you for submitting your manuscript to PLOS ONE. After careful consideration, we feel that it has merit but does not fully meet PLOS ONE’s publication criteria as it currently stands. Therefore, we invite you to submit a revised version of the manuscript that addresses the points raised during the review process.

We look forward to receiving your revised manuscript.

Kind regards,

Clement Ameh Yaro, Ph.D

Academic Editor

PLOS ONE

Journal Requirements:

3.  We note that Figure 1, 4, 5, 6, 7, 8, 9, 10, and 11 in your submission contain [map/satellite] images which may be copyrighted. All PLOS content is published under the Creative Commons Attribution License (CC BY 4.0), which means that the manuscript, images, and Supporting Information files will be freely available online, and any third party is permitted to access, download, copy, distribute, and use these materials in any way, even commercially, with proper attribution. For these reasons, we cannot publish previously copyrighted maps or satellite images created using proprietary data, such as Google software (Google Maps, Street View, and Earth). For more information, see our copyright guidelines: http://journals.plos.org/plosone/s/licenses-and-copyright.

a. You may seek permission from the original copyright holder of Figure 1, 4, 5, 6, 7, 8, 9, 10, and 11 to publish the content specifically under the CC BY 4.0 license.

Reviewers' comments:

Reviewer's Responses to Questions

**Comments to the Author**

1. Is the manuscript technically sound, and do the data support the conclusions?

Reviewer #1: Yes

Reviewer #2: Yes

Reviewer #3: Yes

Reviewer #4: Partly

Reviewer #5: Yes

Reviewer #6: Partly

2. Has the statistical analysis been performed appropriately and rigorously? 

Reviewer #1: Yes

Reviewer #2: Yes

Reviewer #3: Yes

Reviewer #4: Yes

Reviewer #5: Yes

Reviewer #6: Yes

3. Have the authors made all data underlying the findings in their manuscript fully available?

Reviewer #1: Yes

Reviewer #2: Yes

Reviewer #3: No

Reviewer #4: Yes

Reviewer #5: Yes

Reviewer #6: Yes

4. Is the manuscript presented in an intelligible fashion and written in standard English?

Reviewer #1: Yes

Reviewer #2: Yes

Reviewer #3: Yes

Reviewer #4: No

Reviewer #5: Yes

Reviewer #6: Yes

5. Review Comments to the Author

Reviewer #1: This manuscript is very rich in content, with complete data and correct statistical methods. In Nigeria, Water, Sanitation, and Hygiene (WASH) conditions are determined by financial status, level of education, living conditions, and the gender of the household head. Since my research field differs from the content described in this manuscript, some issues need to be adjusted or clarified during the review process.

1.The manuscript determines the size of a household based on the number of household members, with households of more than 6 people classified as large-scale and those with 6 or fewer people as small-scale. Where does the number 6 come from? Is there no category for medium-sized households under this classification?

2.“The proportions were grouped into three equal categories, each category comprising 33.3% of the household population.” What is the specific classification method? Is it to calculate the total score for each household, divide it by the number of household members, rank the households by this number, and then classify the top third as high, the middle third as medium, and the bottom third as low in terms of household education level?

3.The data collected in the manuscript is from 2018, and it is now 2025. Is the time span too large?

Reviewer #2: The manuscript is technically sound and the data support the conclusion. The statistical analysis has been appropriately and rigorously done. The manuscript is written in standard English, and in an intelligible manner. However, the author should check Figure 1. On Figure one , there are two South-West geo-political zones. This should be corrected.

Reviewer #3: The paper is very interesting by providing spatial method in analysing household factors on WASH access. The paper has been written very thorough and complete, but there are some items can be improved as follow:

1.The paper is very long, perhaps can be considered to reduce the word count.

2.Figure 2 and Table 5. Would it be possible to explain what does the values in the table representing for? Is it the numeric score of WASH access and household factors? Is each variable also analysed as binary variable (improved vs unimproved)

3.Figure 3. Is it possible to have the cartogram plotting on the geographical boundaries? It hard to understand which circle belong to which area.

4.Page 19. “However, results show that HBMWQ and HWRR exhibit stronger causal relationships with UN compared to HWLLE and HWRR.” Is UN mean UW?

5.Table 8. UW has been explained in the text. But US and UH seems missing from the explanation

6.Page 21: “This implies that the majority of married couples usually stay married till old age”. IS this relevant to the WASH access?

7.Page 21: “We observe that the distribution of older adults has a similar spatial pattern to that of crime rate and social restiveness in Nigeria. For instance, the North East has been the epicentre of insurgency and banditry for more than a decade and the most vulnerable people are the elderly persons who are physically weak as a result of old age. Whereas in the South-South, the ongoing environmental degradation as a result of oil pollution and the consequent deterioration in public health could result in low life expectancy in oil-producing areas.” This part perhaps need to be discussed closer in later part in 1st paragraph in page 24.

8.Page 21: “We observed that unimproved hygiene is a general but ugly phenomenon that is rampant among people across all socioeconomic strata in Nigeria.” Is there any impact from COVID-19 pandemic massive promotion of hand hygiene to the improvement of hygiene practice in Nigeria?

9.Page 22: “Such suspected influencing factors could be more environmental conditions rather than household factors.” Beside environmental factors, local state policy and programs, experience with WASH program in particular state may also influence WASH access and behaviour.

10.Page 26: “In this study, we present an elite analytical method that not only establishes a causal relationship between household factors and WASH conditions but also models the spatial variability of the causal relationship as well as the spatial pattern of individual household parameter influence on WASH conditions.” Would you please elaborate more, what tis the new perspective or understanding can be contributed using these spatial analytical methods compare to standards statistical analysis?

11.In discussion section there are some redundancies in explaining influence each household factors (FHH, economic, education etc factors). It could be reorganised to have more concise explanation.

12.Conclusion has been made based on the findings, but perhaps conclusion on the benefit of using the spatial analysis is worth to be included.

13.There are many studies have similar findings in regards to the household’s factors influence of WASH access which can be refer in discussion section, which is currently lacking

Reviewer #4: Clarity on "Unimproved Hygiene": The manuscript frequently mentions "unimproved hygiene." Please define this term clearly in the introduction or methods section. Specifying what constitutes "unimproved hygiene" will help readers better understand the scope of the study.

Justification for Statistical Methods: While the methods are mentioned, provide a brief justification for using spatial statistics in addition to non-spatial methods. Why is a spatial approach particularly valuable for this research question? Expand on why spatial regression was appropriate.

Interpretation of DoD: Expand on the interpretation of the Degree of Dependence (DoD_j) values. While the results state that residency type has a higher DoD_j for water source choice, explain the practical implications of this finding. What does a DoD_j of 0.998 mean in terms of policy or interventions?

Limitations: The authors acknowledge that the best predictive models only account for 76% of the variance. Elaborate on the potential factors that might explain the remaining variance and discuss the implications of these unmeasured factors. What specific data limitations might affect the conclusions?

English Language and Grammar: While generally well-written, the manuscript could benefit from a careful review by a native English speaker to address minor grammatical issues and improve clarity. For example, in the abstract, "...morbidity of school-age children" could be rephrased for smoother reading. Also, check for consistency in terminology.

Policy Recommendations: While the policy recommendations are relevant, make them more specific and actionable. For example, instead of just stating "the need for more focused policy action," suggest specific types of interventions, target populations, or potential partnerships.

Literature Review: Consider expanding the literature review to include more recent studies on WASH interventions in Nigeria and similar contexts. This will help contextualize the study's findings and demonstrate its contribution to the existing body of knowledge.

Reviewer #5: The abstract of this paper can be shortened. The literature gap is not evident in the introduction section. I advise author to elaborate his contribution in literature. I suggest adding a table, describing the variables used in this research along with their descriptive statistics (in variable measurement section or appendices). Autor should elaborate the choice of using techniques like Global regression models, OLS, Local Regression model and Multiscale geographically weighted regression, Thier importance in the context of this research. Inconsistence in table notes, a few tables in the manuscript lack table note for example 7 and 8. The discussion section is well written, but conclusion of this research is too short, I recommend author to rewrite conclusion of this manuscript by taking research practice in consideration.

Reviewer #6: Line numbers should be added for providing comments line by line from reviewers.

Sample design: Instead of referring sampling procedure in the DHS report, it is better to describe the type of sampling at the cluster and household levels.

How does a protected spring differ from an unprotected spring?

For education, does low education include primary and no education?

How to define household size by 6? Is there any reason for using 6 as the cutoff point?

In Table 4, adjusted R2 and AIC should be described separately for model fitness because lower AIC values indicate better fitness. In this analysis, MGWR yields better model statistics because of higher adjusted R2 and lower AIC.

In the Local Regression Model, the Authors added Structural Equation Modelling (SEM), and Special Lag Models (SLM). What findings represent SEM in your findings? Do you mean it as a Spatial Error Model?

Another one is that it is unclear whether Spatial or Special Lag Models exist. Please answer thoroughly for clarification.

In supporting information 1, binary logistic regression was shown where independent variables were mentioned as household characteristics. Under the analytical procedure (page 7), household factors are HMWQ, HLLE, HRR, and FHH. Therefore, these definitions are not clearly cut.

Another question is why regression analysis at the individual level is added in supporting information. Your study objective is to examine the relationship between household factors and WASH in terms of spatial context.

For Geospatial analysis, the data set should be representative of the entire population like Census data. In the DHS data, the sampling is a probability proportionate method and so although it is a nationwide survey, the target population is not covered for geospatial analysis. Therefore, authors are suggested to add this limitation as a discussion point.

The conclusion should be based on the research findings.

it is suggested to modify the statement “Households with these characteristics may not be mentally, physically, and economically prepared to provide improved WASH for the members due to a possible lack of information about the health consequences, financial incapacity and poverty of the environment in which they live” because this study does not include analysis on mental status and access to information about the health consequences.

6. PLOS authors have the option to publish the peer review history of their article (what does this mean? ). If published, this will include your full peer review and any attached files.

**Do you want your identity to be public for this peer review?** For information about this choice, including consent withdrawal, please see our Privacy Policy .

Reviewer #1: No

Reviewer #2: No

Reviewer #3: **Yes: ** Ni Made Utami Dwipayanti

Reviewer #4: **Yes: ** Hala Awad

Reviewer #5: **Yes: ** S M NABEEL UL HAQ

Reviewer #6: **Yes: ** Dr. May Soe Aung, Associate Professor, Department of Preventive and Social Medicine, University of Medicine (1), Yangon, Myanmar

---

## [Author Response · Author response to Decision Letter 1]

20 May 2025

Response: The manuscript has been formatted to meet PLOS ONE’s style requirements, including those for file naming.

Response: Ethics statement has been included in the method section

3. We note that Figure 1, 4, 5, 6, 7, 8, 9, 10, and 11 in your submission contain [map/satellite] images which may be copyrighted. All PLOS content is published under the Creative Commons Attribution License (CC BY 4.0), which means that the manuscript, images, and Supporting Information files will be freely available online, and any third party is permitted to access, download, copy, distribute, and use these materials in any way, even commercially, with proper attribution. For these reasons, we cannot publish previously copyrighted maps or satellite images created using proprietary data, such as Google software (Google Maps, Street View, and Earth). For more information, see our copyright guidelines: http://journals.plos.org/plosone/s/licenses-and-copyright.

Response: All figures were originally generated by the authors and none of them have been copyrighted. We do not need to take permission from anyone. All the figures and tables presented in the manuscript are genuine, original and none of them have been published, used or appeared in any form. The figures are exclusively generated for this study. The figures are geospatial expressions of the analyses results.

Response: The Supporting Information files have been listed

Response: We have ensured that only cited references are listed. None of the cited papers has been retracted so far. There is no significant change to the reference list. We only introduced two additional references

Final Remarks: All the concerns of all the reviewers have been comprehensively addressed

---

## [Decision Letter · Decision Letter 1]

19 Jun 2025

PONE-D-24-54290R1Geospatial Assessment of Household Water, Sanitation and Hygiene Conditions and Associated Factors in Nigeria: A Causal Relationship ModelPLOS ONE

Dear Dr. Akinwumiju,

Thank you for submitting your manuscript to PLOS ONE. After careful consideration, we feel that it has merit but does not fully meet PLOS ONE’s publication criteria as it currently stands. Therefore, we invite you to submit a revised version of the manuscript that addresses the points raised during the review process.

We look forward to receiving your revised manuscript.

Kind regards,

Clement Ameh Yaro, Ph.D

Academic Editor

PLOS ONE

Journal Requirements:

Reviewers' comments:

Reviewer's Responses to Questions

**Comments to the Author**

1. If the authors have adequately addressed your comments raised in a previous round of review and you feel that this manuscript is now acceptable for publication, you may indicate that here to bypass the “Comments to the Author” section, enter your conflict of interest statement in the “Confidential to Editor” section, and submit your "Accept" recommendation.

Reviewer #1: All comments have been addressed

Reviewer #2: All comments have been addressed

Reviewer #3: All comments have been addressed

Reviewer #4: (No Response)

2. Is the manuscript technically sound, and do the data support the conclusions?

Reviewer #1: Yes

Reviewer #2: Yes

Reviewer #3: Yes

Reviewer #4: Yes

3. Has the statistical analysis been performed appropriately and rigorously? 

Reviewer #1: Yes

Reviewer #2: Yes

Reviewer #3: Yes

Reviewer #4: Yes

4. Have the authors made all data underlying the findings in their manuscript fully available?

Reviewer #1: Yes

Reviewer #2: Yes

Reviewer #3: Yes

Reviewer #4: Yes

5. Is the manuscript presented in an intelligible fashion and written in standard English?

Reviewer #1: Yes

Reviewer #2: Yes

Reviewer #3: Yes

Reviewer #4: Yes

6. Review Comments to the Author

Reviewer #1: (No Response)

Reviewer #2: (No Response)

Reviewer #3: Dear Author,

Thank you for addressing all my comments in revise the manuscript accordingly.

regards

Reviewer #4: 1. Scientific Rigor and Statistical Analysis

The research utilizes both spatial and non-spatial statistical techniques, such as spatial regression and Degree of Dependence (DoD) calculations, to thoroughly examine the connections between household factors and WASH conditions. The statistical methods are suitable and meticulously implemented, including clear documentation of model fit indices (DoD, Adjusted R², AICc) and levels of significance. Leveraging a large, nationally representative dataset (DHS) guarantees an adequate sample size and the ability to replicate findings across various household and geographic settings. The conclusions drawn are strongly backed by the data and analyses provided.

2. Data Availability

The manuscript adheres completely to the PLOS Data Policy. All pertinent datasets are accessible without restrictions, both in the manuscript and its Supporting Information files, as well as through public repositories (Google Drive link and the DHS Program website). The data is anonymized to safeguard participant privacy, with explicit statements outlining the management of sensitive information.

3. Ethics and Research Integrity

The Methods section contains a suitable ethics statement that outlines the utilization of DHS data, IRB approval, and the processes for anonymization that safeguard the confidentiality of respondents. Based on the information provided, there are no issues related to research ethics, dual publication, or publication ethics.

4. Presentation and Language

The manuscript is mostly clear, well-composed, and easy to understand. The scientific terminology used is standard and clear, allowing a wide readership to grasp the findings. There are some minor typographical and grammatical mistakes that need correction for better clarity (such as sentence construction in the abstract, pluralization issues, and formal wording in response letters). These issues are relatively minor and do not hinder comprehension.

5. Figures, Tables, and Supporting Information

All figures and tables are original creations by the authors, ensuring they are free from copyright issues. Supporting information is meticulously listed and referenced according to journal guidelines.

Additional Comments/Suggestions

• Kindly examine the manuscript for minor typographical and grammatical errors, particularly in the abstract and the response to reviewers, to improve clarity further.

• Confirm that all supporting information files are adequately captioned and cited in the main text, following PLOS ONE guidelines.

• The reference list is current and does not include any retracted articles, with new references properly introduced.

7. PLOS authors have the option to publish the peer review history of their article (what does this mean? ). If published, this will include your full peer review and any attached files.

**Do you want your identity to be public for this peer review?** For information about this choice, including consent withdrawal, please see our Privacy Policy .

Reviewer #1: No

Reviewer #2: No

Reviewer #3: **Yes: ** Ni Made Utami Dwipayanti

Reviewer #4: **Yes: ** Hala Awad Ahmed

---

## [Author Response · Author response to Decision Letter 2]

11 Jul 2025

Response to Reviewers

1. All typos and grammatical errors have been taken care of. We have improved on the grammar and the quality of the entire manuscript.

2. In line with PLOS ONE guidelines, all supporting information files are adequately captioned and cited in the main text.

Response to Editor

1. The manuscript has been formatted to meet PLOS ONE’s style requirements, including those for file naming.

2. We have ensured that only cited references are listed. None of the cited papers has been retracted so far. There is no significant change to the reference list.

---

## [Decision Letter · Decision Letter 2]

29 Jul 2025

Geospatial Assessment of Household Water, Sanitation and Hygiene Conditions and Associated Factors in Nigeria: A Causal Relationship Model

PONE-D-24-54290R2

Dear Dr. Akinwumiju,

We’re pleased to inform you that your manuscript has been judged scientifically suitable for publication and will be formally accepted for publication once it meets all outstanding technical requirements.

Kind regards,

Clement Ameh Yaro, Ph.D

Academic Editor

PLOS ONE

Additional Editor Comments (optional):

Reviewers' comments:

Reviewer's Responses to Questions

**Comments to the Author**

1. If the authors have adequately addressed your comments raised in a previous round of review and you feel that this manuscript is now acceptable for publication, you may indicate that here to bypass the “Comments to the Author” section, enter your conflict of interest statement in the “Confidential to Editor” section, and submit your "Accept" recommendation.

Reviewer #4: (No Response)

2. Is the manuscript technically sound, and do the data support the conclusions?

Reviewer #4: Yes

3. Has the statistical analysis been performed appropriately and rigorously? 

Reviewer #4: Yes

4. Have the authors made all data underlying the findings in their manuscript fully available?

Reviewer #4: Yes

5. Is the manuscript presented in an intelligible fashion and written in standard English?

Reviewer #4: Yes

6. Review Comments to the Author

Reviewer #4: The revised manuscript titled "Geospatial Assessment of Household Water, Sanitation, and Hygiene Conditions and Associated Factors in Nigeria: A Causal Relationship Model" is well-structured and clearly written. The authors have thoroughly addressed previous reviewer comments, improved the grammar and clarity of the text, and ensured full adherence to PLOS ONE’s submission guidelines.

The study design is technically sound, with robust use of spatial and non-spatial statistical analyses. The authors have justified their methodological choices, including the application of Ordinary Least Squares (OLS) and Multiscale Geographically Weighted Regression (MGWR) models, which are appropriate for the research objectives. The data support the conclusions, and the findings are meaningful and relevant, especially for policy implications in low- and middle-income countries like Nigeria.

Statistical analyses were performed rigorously, with clear presentation of model diagnostics, coefficient interpretation, and adjusted R² and AIC values. The spatial heterogeneity of WASH conditions and household factors is well illustrated through the use of GIS and thematic cartograms.

The manuscript also meets data transparency requirements. The datasets are publicly available via OSF and the DHS Program website, and the authors have provided a clear Data Availability Statement.

Overall, this is a high-quality manuscript that offers a valuable contribution to the literature on WASH inequalities and spatial health analysis. I recommend acceptance for publication.

7. PLOS authors have the option to publish the peer review history of their article (what does this mean? ). If published, this will include your full peer review and any attached files.

**Do you want your identity to be public for this peer review?** For information about this choice, including consent withdrawal, please see our Privacy Policy .

Reviewer #4: **Yes: ** Hala Awad Ahmed

---

## [Editor Report · Acceptance letter]

PONE-D-24-54290R2

PLOS ONE

Dear Dr. Akinwumiju,

I'm pleased to inform you that your manuscript has been deemed suitable for publication in PLOS ONE. Congratulations! Your manuscript is now being handed over to our production team.

Kind regards,

on behalf of

Dr. Clement Ameh Yaro

Academic Editor

PLOS ONE